# Inter-Blade Vortex and Vortex Rope Characteristics of a Pump-Turbine in Turbine Mode under Low Flow Rate Conditions

**Seung-Jun Kim [1,2], Jun-Won Suh [2], Young-Seok Choi [1,2], Jungwan Park [3], No-Hyun Park [4] and Jin-Hyuk Kim [1,2,\*]**

[1]  Industrial Technology (Green Process and Energy System Engineering), Korea University of Science & Technology, Daejeon 34113, Korea; kimsj617@kitech.re.kr (S.-J.K.); yschoi@kitech.re.kr (Y.-S.C.)
[2]  Thermal & Fluid System R&D Group, Korea Institute of Industrial Technology, Cheonan 31056, Korea; taylor881204@kitech.re.kr
[3]  Hydropower Design & Technology Group, Korea Hydro & Nuclear Power Co. Ltd, Daejeon 34101, Korea; park.jungwan@khnp.co.kr
[4]  Kumsung Engineering & Construction Co. Ltd, Nonsan 33020, Korea; nhpark@ekcf.co.kr
\*  Correspondence: jinhyuk@kitech.re.kr; Tel.: +82-41-589-8447

**Abstract:** Pump-turbines are often used to provide a stable power supply with a constant frequency in response to intermittent renewable energy resources. However, existing pumped-storage power stations often operate under off-design conditions because of the increasing amounts of inconsistent renewable resources that have been added to the grid. Under off-design low flow rate conditions, inter-blade vortex and vortex rope phenomena usually develop in the runner and draft tube passages, respectively, in turbine mode. These vortices cause complicated flow patterns and pressure fluctuations that destabilize the operation of the pump-turbine system. Therefore, this study investigates the influence of correlation between the inter-blade vortex and vortex rope phenomena under low flow rate conditions. Three-dimensional steady- and unsteady-state Reynolds-averaged Navier–Stokes equations were calculated with a two-phase flow analysis using a shear stress transport as the turbulence model. The inter-blade vortices in the runner passages were captured well at the low flow rate conditions, and the vortex rope was found to develop within a specific range of low flow rates. These vortex regions showed a blockage effect and complicated flow characteristics with backflow in the passages. Moreover, higher unsteady pressure characteristics occurred at locations where the vortices were especially pronounced.

**Keywords:** pump-turbine; turbine mode; inter-blade vortex; vortex rope; internal flow characteristics; unsteady pressure

## 1. Introduction

Pumped-storage hydroelectric power stations generally operate by transferring water from the lower to the upper reservoir using idle power in pump mode during light load times (low electrical power demand), and after saving energy, discharging water in turbine mode during peak load times (high electrical power demand). In addition, such power stations typically have operation characteristics that enable rapid deployment of emergency power with fast start-up and quick load increase and decrease capabilities [1].

Pumped-storage power stations are now often used to provide a stable power supply with a constant frequency for responding to fluctuations in renewable energy resources, such as solar and wind power, which have been increasingly added to the grid over the last several decades. The

amount of power generated by these renewable energy resources is not constant, causing larger prediction errors relative to power generation from other resources. Such large prediction errors result in differences between the planned and actual power supply, and thereby require more frequent load increase and decrease cycles to accommodate the power fluctuations from the inconsistent resources. Moreover, existing pumped storage power stations often operate at flow rates that are either higher or lower than the design conditions, resulting in emergency stops, failures, and reduced service lifetimes of the pump-turbine systems [2–4].

Under low flow rate conditions, vortices with complicated internal flow patterns can be generated in the passage of the pump-turbine's runner and draft tube during the turbine mode. The inter-blade vortex in the runner passages and the vortex rope in the cone of draft tube not only reduce the system's hydraulic performance but also induce unsteady pressure pulsations, which can result in destabilizing vibrations and noise. The characteristics of these vortices must be elucidated to expand the stable operating range of pump-turbine systems and thereby ensure a consistent power supply [5].

The inter-blade vortex and vortex rope in the hydro turbines have common characteristics with the vortex characteristics in the pump-turbines. Yamamoto et al. [6] studied to reveal a physical mechanism for the inter-blade vortex development of a Francis turbine at deep part load conditions by the numerical simulation. Magnoli [7] conducted numerical analyses of the Francis turbine at the different load conditions by measuring runner blade pressure fluctuations with the rotor-stator interaction, rotating vortex rope, and the inter-blade vortex. Fay [8] investigated the low-frequency in the Francis turbine using transient torque equation with the rotor-stator interaction, inter-blade vortices and spiraling vortex flow as the various potential pulsation sources.

In related studies on the flow characteristics of a pump-turbine at off-design conditions, Li et al. [9] conducted numerical analyses of the flow structures in a reversible pump-turbine model, focusing on the hydraulic force of the impeller in generating mode; they found that the dominant force components depend on the working conditions. Hasmatuchi et al. [10] studied experimentally using a reduced-scale pump-turbine model in turbine mode to investigate rotating stall by measuring pressure fluctuations in the wicket gate channels synchronized using the high-speed flow visualization under off-design conditions. Yan et al. [11] conducted unsteady state incompressible analyses on a reduced-scale radial pump-turbine model in turbine mode at off-design conditions to predict the amplitude of the normal forces on the impeller. Liao et al. [12] investigated pressure pulsations in the small-opening operation mode of a pump-turbine, with and without consideration of the weak compressibility of water, to observe the internal flow and pressure characteristics in the runner passages and draft tube.

Among the studies of pump-turbines under off-design conditions, several studies have investigated the vortex characteristics in the draft tube. Wang et al. [13] carried out the unsteady state analysis to investigate the flow in straight- and elbow-type draft tubes of a pump-turbine under partial load, with both a very large eddy simulation and a shear stress transport (SST) Reynolds-averaged Navier-Stokes (RANS) model. They also studied the influence of a water jet introduced to suppress the vortex rope. Kirschner et al. [14] conducted an experimental observation with visualization of the vortex phenomena and a numerical investigation of the pressure fluctuations and radial forces on the wall of draft tubein a pump-turbine model under various flow conditions. Lai et al. [15] experimentally investigated the velocity distributions in the draft tube of a pump-turbine model by laser Doppler velocimetry with visualization of the vortex rope. They revealed the flow features in the cone of draft tube and the distributions of axial and tangential velocities under different operating conditions. Thus, several studies have been conducted on the internal flow and pressure characteristics in the pump-turbine and the vortex rope characteristics in the draft tube under low flow rate conditions in turbine mode. However, under the low flow rate conditions where vortex rope occurs in the draft tube, the inter-blade vortices can be also occurred in the runner passages simultaneously and then it is directly influenced on the vortex rope; these vortex characteristics according to the flow rate and the relationship between the inter-blade vortex and vortex rope characteristics in turbine mode of the pump-turbine have not yet been detailed under low flow rate conditions. These vortices can adversely

affect the pump-turbine system with complicated internal flow and unsteady pressure characteristics. Therefore, the understanding in cause and relationship of the vortex characteristics is essential to ensure stable operations under off-design low flow rate conditions.

In this study, three-dimensional (3D) steady- and unsteady-state RANS analyses were conducted using a SST turbulence model with two-phase flow analysis to investigate the characteristics of correlations between the vortices in the passage of the runner and draft tube. The numerical analyses compared the best efficiency point (BEP) condition with various low flow rate conditions according to the guide vane angle (GVA). The flow structures, correlations, and unsteady pressure characteristics of the inter-blade vortex and vortex rope were investigated according to flow rate, and the swirl number at the outlet of runner was applied to observe the flow characteristics in the presence of a vortex rope in the draft tube of the pump-turbine.

## 2. Pump-Turbine (Turbine Mode)

In this study, 3D steady- and unsteady-state numerical analyses of a pump-turbine were conducted in turbine mode with a specific speed $N_S$ of 120 m·kW·min$^{-1}$. Equation (1) was used to calculate the specific speed. Figure 1 shows the 3D model and meridional plane of the system's main components with the flow direction in turbine mode. The energy coefficient $E_{nD}$, discharge coefficient $Q_{nD}$, and speed factor $n_{ED}$ of the system are listed in Table 1 and were derived using equations defined by the IEC 60193 standard, along with Equations (2)–(4) [16].

$$N_s = \frac{N\sqrt{P}}{H^{\frac{5}{4}}} \tag{1}$$

$$E_{nD} = \frac{E}{n^2 D^2} \tag{2}$$

$$Q_{nD} = \frac{Q}{nD^3} \tag{3}$$

$$n_{ED} = \frac{nD}{\sqrt{gH}} \tag{4}$$

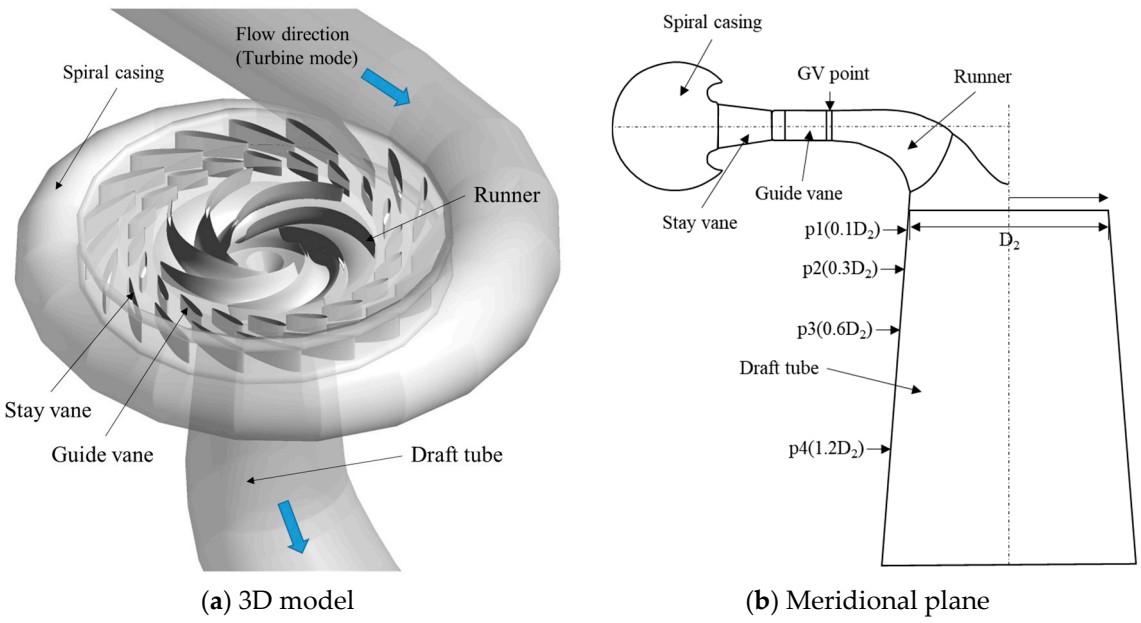

(**a**) 3D model　　　　　　　　　　　　(**b**) Meridional plane

**Figure 1.** (**a**) 3D model and (**b**) meridional plane of the pump-turbine.

**Table 1.** Specifications of the pump-turbine (in turbine mode).

| Specifications | Values |
|---|---|
| Specific speed, $N_s$ (m·kW·min$^{-1}$) | 120 |
| Runner outlet diameter, $D_2$ (m) | 2.78 |
| Energy coefficient, $E_{nD}$ (-) | 14.91 |
| Discharge coefficient, $Q_{nD}$ (-) | 0.81 |
| Speed factor, $n_{ED}$ (-) | 0.25 |
| No. of runner blade | 7 |
| No. of stay vane | 20 |
| No. of guide vane | 20 |

## 3. Numerical Analysis Methods

This study used ANSYS CFX-18.2 commercial software (ANSYS Inc., Canonsburg, PA, USA) for the steady- and unsteady-state analyses of the pump-turbine's incompressible internal flow field [17]. In a package of ANSYS CFX, Turbo-Grid was used to generate the numerical grids for the runner and guide vane, and the spiral casing, stay vane, and draft tube, were generated using ICEM-CFD. The boundary conditions for the numerical analyses were set using CFX-Pre; the solving governing equations and posting the results were conducted using CFX-Solver and CFX-Post, respectively. The steady- and unsteady-state RANS equations for the incompressible flow characteristics in the pump-turbine were calculated using the governing equations, which were discretized with the finite volume method. The discretizations of the advection and transient scheme were solved with the high resolution and second order backward Euler schemes to ensure the physical boundaries.

Numerical analyses of fluid machines that rotate in the axial direction and have an axisymmetric structure are generally performed on the period corresponding to the passage of one blade to decrease the computational cost and improve convergence. However, the flow areas of the spiral casing in a pump-turbine are gradually reduced from the inlet in turbine mode, and the dynamics of the internal flow are generated between the stay vane and the shape of the casing tongue. Therefore, axisymmetric periodic conditions could not be applied in this study, and this analysis was conducted over all stay vane, guide vane, and runner blade components [18].

Water and vapor at 25 °C were considered as the working fluids in a two-phase flow to account for cavitation characteristics with the Rayleigh Plesset cavitation model, which describes the growth and collapsing of vapor bubbles in a liquid as the homogeneous model [19]. The mean diameter of the bubble was set to $2.0 \times 10^{-6}$ and the water saturation pressure was set to 3169.9 Pa.

Figure 2 shows the numerical grids of the pump-turbine. Tetrahedral grids were constructed in the computational domains of the spiral casing and stay vanes, and hexahedral grids were constructed in other regions, such as the guide vane, runner blade, and draft tube. O-type grids were used near the runner blade surface, which has a relatively complicated flow structure, to apply wall function processing conditions with $y^+ \leq 100$ as the first grid point [17].

Figure 3 shows a numerical grid dependency test; the test results show the normalized efficiency according to the node number. The different girds were applied by changing the nodes while keeping the node density ratio for each component about the total nodes and $y^+$ on the runner blade surface. Node numbers in the range of $2.6-9.8 \times 10^6$ were compared, and finally the optimum node number was found to be $7.49 \times 10^6$. The number of nodes for each component of the optimum grid was listed in Table 2.

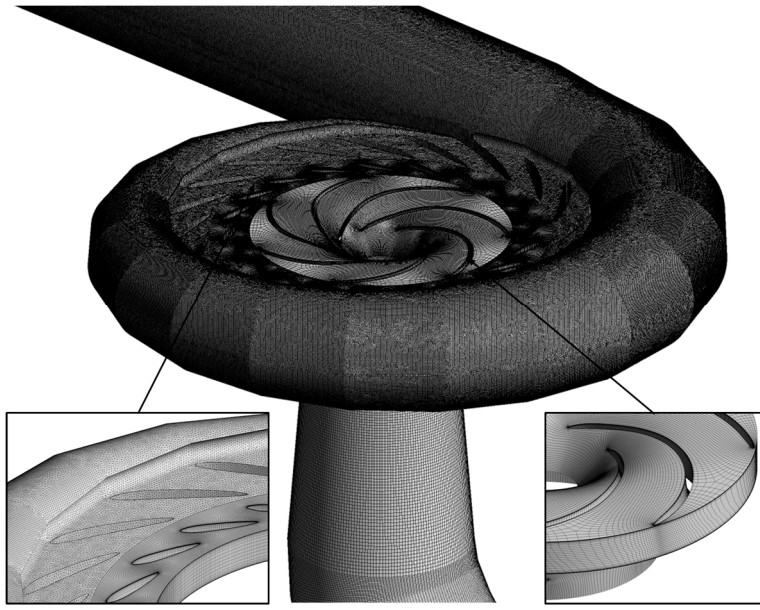

**Figure 2.** Schematic of the numerical grids in the pump-turbine.

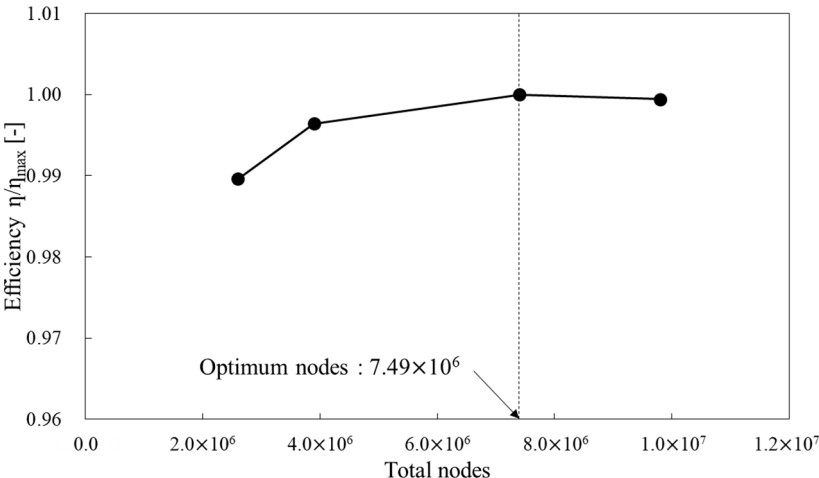

**Figure 3.** Numerical grid dependency test results.

**Table 2.** Number of nodes for the optimum numerical grid.

| Components | No. Nodes | Grid Type |
|---|---|---|
| Spiral casing + Stay vane | $2.84 \times 10^6$ | Unstructured |
| Guide vane | $1.52 \times 10^6$ | Structured |
| Runner | $1.47 \times 10^6$ | Structured |
| Draft tube | $1.66 \times 10^6$ | Structured |
| Total nodes | $7.49 \times 10^6$ | - |

As the boundary conditions, the total pressure and static pressure were set accounting for the water level of the upper and lower reservoirs in the turbine mode at the pump-turbine's inlet and outlet, respectively. The SST turbulence model was used, which is suitable for precise predictions of the flow separation in an adverse pressure gradient [20,21]. This model employs k-ω and k-ε models according to the distance from the wall respectively, and the blending function between these two models ensures the smooth transitions [17]. For the steady-state analyses, stage boundary conditions were applied to connect the surfaces between the stationary and rotating domains; for the unsteady-state analyses, transient-rotor-stator boundary conditions were applied. In order to improve

convergence and decrease the computational time of the unsteady-state analyses, the steady-state analysis results were used as the initial value. The resolution during the unsteady-state analysis was the runner revolution of 3° per time step. Widmer et al. [1] found good agreement regarding local vortex formation, pressure distribution, and width and frequency of the rotating stall at resolutions between 1° and 5° per time step. Therefore, in this study, the time step was set to 0.001389 s, with a total time of 0.8335 s, and the unsteady-state analysis was performed over a total of five revolutions of the runner. To improve convergence, the number of iterations as the loops coefficient was set to five.

Figure 4 shows the measuring points established to investigate the unsteady pressure characteristics according to the internal flow and vortex characteristics. The measuring points were established at regular intervals at the guide vane outlet and the draft tube cone. The location of measuring points at the guide vane outlet was 1.795 $D_2$, where $D_2$ is defined as a reference diameter of the pump-turbine in this study as shown in Figure 1. In this study, the unsteady pressures were measured under various low flow rate conditions at the measuring points marked GV_01, GV_07, and GV_14 of the guide vane outlet and p1–p4 (0.1 $D_2$–1.2 $D_2$ as shown in Figure 1) of the draft tube cone located at L1 as marked on Figure 4.

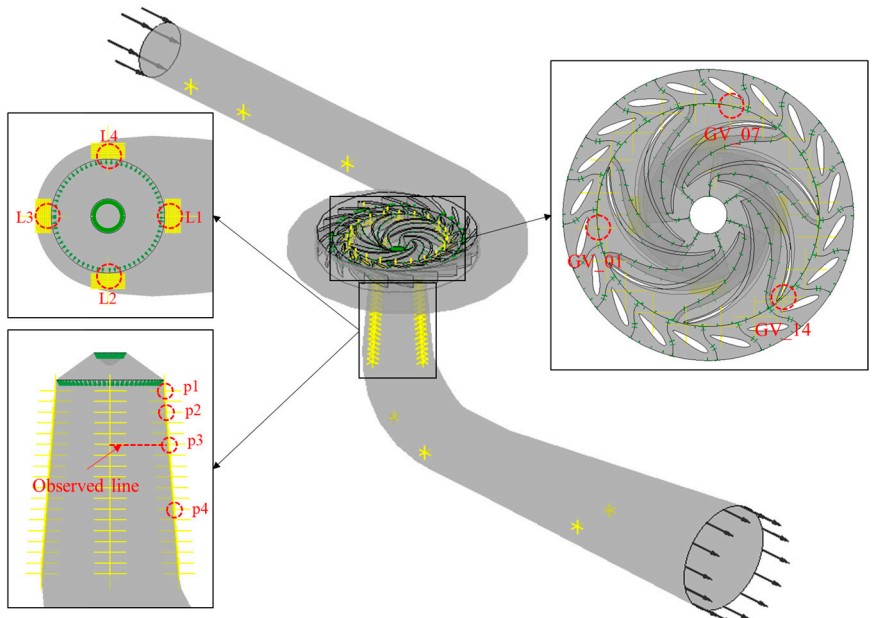

**Figure 4.** Pressure measuring points at the pump-turbine's guide vane and draft tube.

## 4. Results

### 4.1. Validation Test for Results of the Numerical Analyses

To validate the numerical analysis results, Figure 5 compares the steady- and unsteady-state RANS analysis results with the experimental results. The experimental results of the performance test were provided from the real site which is one of the hydro power plants of Korea; and the uncertainty for the power and efficiency were ±1.14% and 0.91%, respectively, as shown in Figure 5 [22]. In the performance curves, POW and EFF indicate the power and efficiency, respectively. The power, efficiency and flow rate related to the validation were normalized by the corresponding values of the BEF from the unsteady state analysis results, respectively. The efficiency and power were calculated by Equations (5) and (6).

$$\eta = \frac{P}{\rho g H Q} \tag{5}$$

$$P = T\omega \tag{6}$$

The unsteady state analysis results were averaged with values during the last revolution of the runner. The trends in the normalized performance curves are generally in good agreement for the results of the unsteady state analysis. Therefore, the results of the numerical analysis were considered valid.

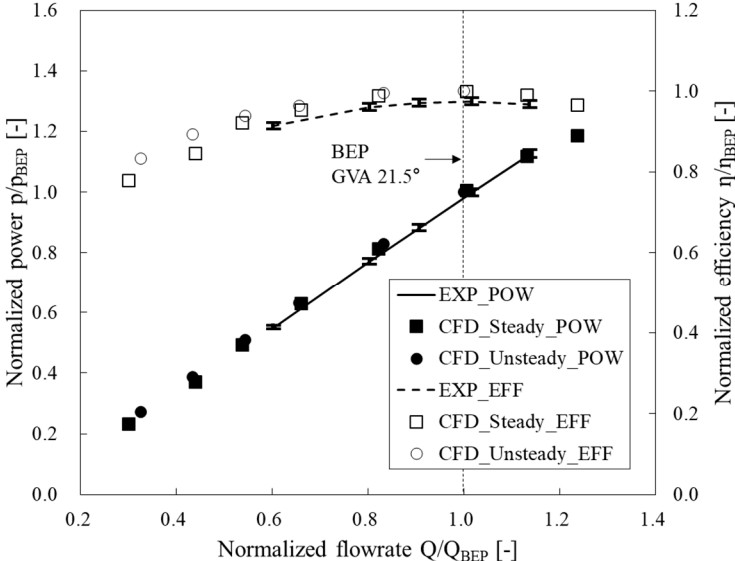

**Figure 5.** Comparison of numerical and experimental performance curves for the pump-turbine.

### 4.2. Inter-Blade Vortex Characteristics According to Flow Rate

Figure 6 shows the internal flow characteristics of runner blades with streamline and iso-surface distributions, which are the time-averaged values of the velocities from the unsteady-state analyses. The velocity in the streamline was normalized by the maximum velocity (80 m/s) of the entire domain. The iso-surface distributions show lower-velocity regions as about 5% of the maximum velocity that can be regarded as a flow stagnation region. Figure 6a shows the BEP condition, characterized by the smooth internal flow along the runner passage, with no developing vortices or flow stagnation regions. In contrast, Figure 6b–f show that vortices were gradually generated in the runner passages as the flow rate decreased, and these vortices caused the flow stagnation regions shown in the iso-surface distributions. Furthermore, more flow stagnation regions due to the vortices in the runner passages developed from the hub to the mid-span than near the shroud span. The volumes of the iso-surface as the stagnation regions (Figure 6) were compared as shown in Figure 7. The normalized volume (stagnation region) was increased gradually as the GVA decreased; however, the area was decreased from the GVA of 7°. It can be seen that the stagnation region induced by the inter-blade vortex was formed near leading edge at the GVA of 7°, therefore the stagnation region was formed in the runner passages relatively smaller than the GVA of 8.5°; this reason can be seen the difference of incidence angle (Figure 8).

To investigate the cause of the vortex development and flow stagnation regions in the runner passages, velocity triangle distributions were compared with the averaged values of the each component of the velocity triangle from hub to shroud of the runner as shown in Figure 8. In the velocity triangle distributions, the black, red, and blue lines show the GVAs of 21.5°, 12.5°, and 7°, respectively. Here, $U$, $W$, and $C$ are the rotational velocity, the relative velocity, and the absolute velocity, respectively; and $C_m$ is the meridional velocity, which represents the flow rate. For the GVAs of 12.5° and 7°, $C_m$ components decreased as flow rate decreased, and the angle of the relative velocity, $\beta$ also changed because the relative velocity angles depend on $C_m$ relative to the $\beta$ of the GVA of 21.5°. Therefore, the

incidence angle, which is different from the blade angle and flow angle, did not correspond with the blade angle as it did under the BEP condition. The change in the flow characteristics in the runner passages in response to the changing incidence angle and flow rate caused the vortices to develop, along with the flow stagnation and complicated flow regions.

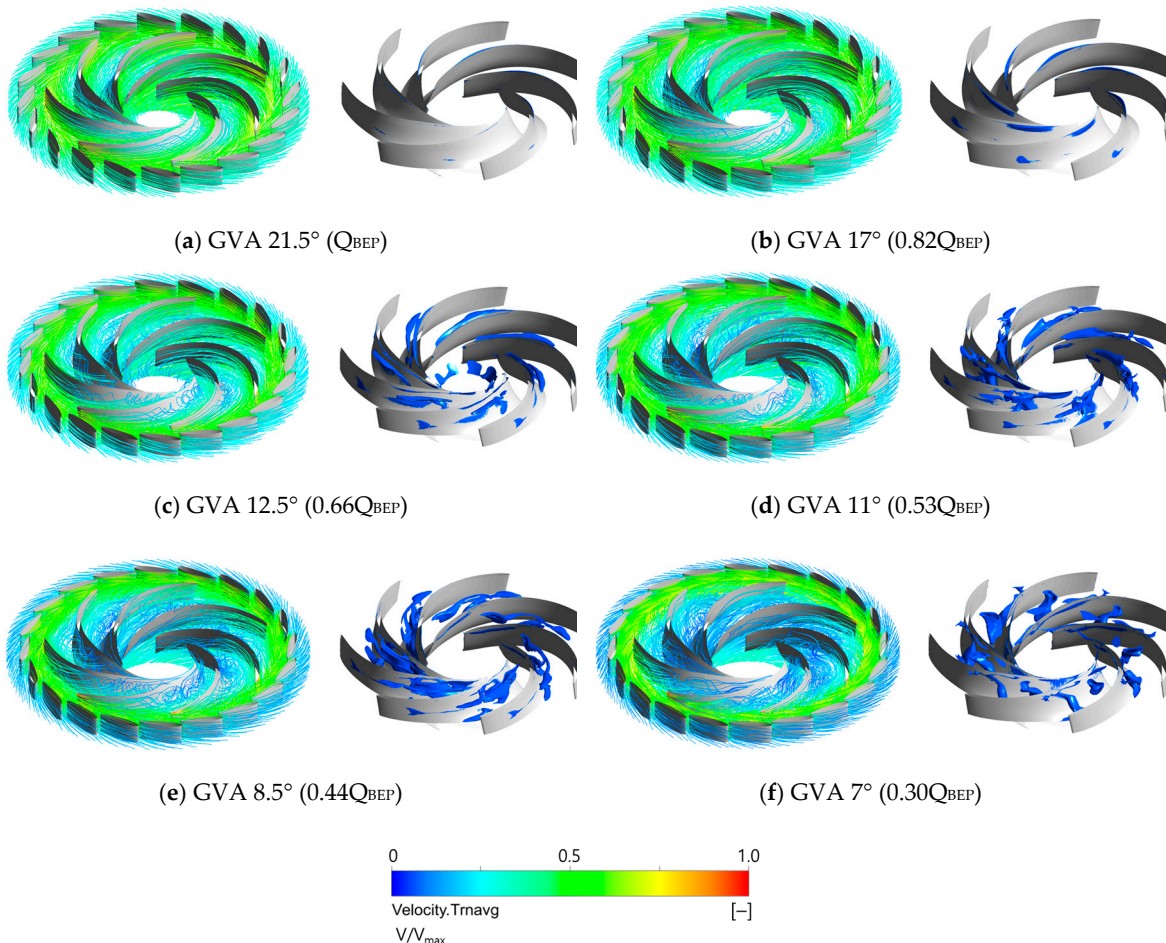

(**a**) GVA 21.5° ($Q_{BEP}$)

(**b**) GVA 17° ($0.82Q_{BEP}$)

(**c**) GVA 12.5° ($0.66Q_{BEP}$)

(**d**) GVA 11° ($0.53Q_{BEP}$)

(**e**) GVA 8.5° ($0.44Q_{BEP}$)

(**f**) GVA 7° ($0.30Q_{BEP}$)

**Figure 6.** Streamline (left) and iso-surface (right) distributions in passages of the runner blade with guide vane angles (GVAs) of (**a**) 21.5°, (**b**) 17°, (**c**) 12.5°, (**d**) 11°, (**e**) 8.5°, and (**f**) 7°.

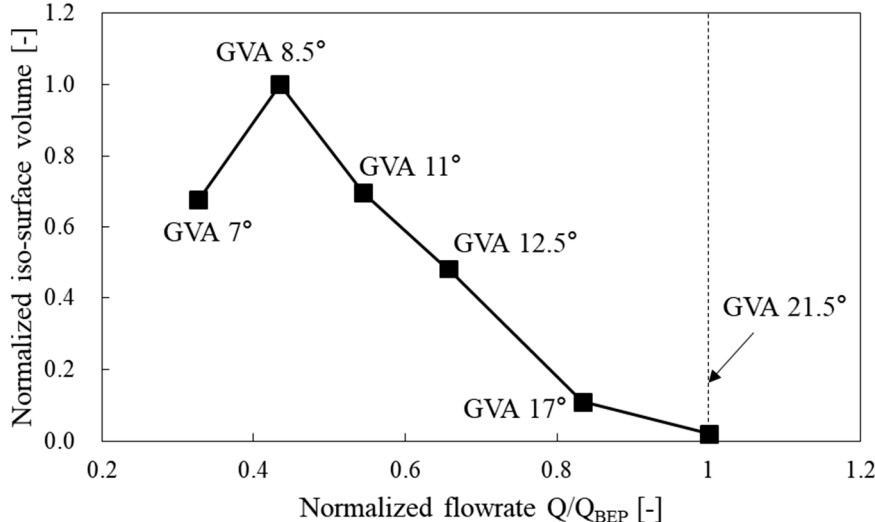

**Figure 7.** Volume distribution of the iso-surface of lower-velocity regions (Figure 6) with GVAs.

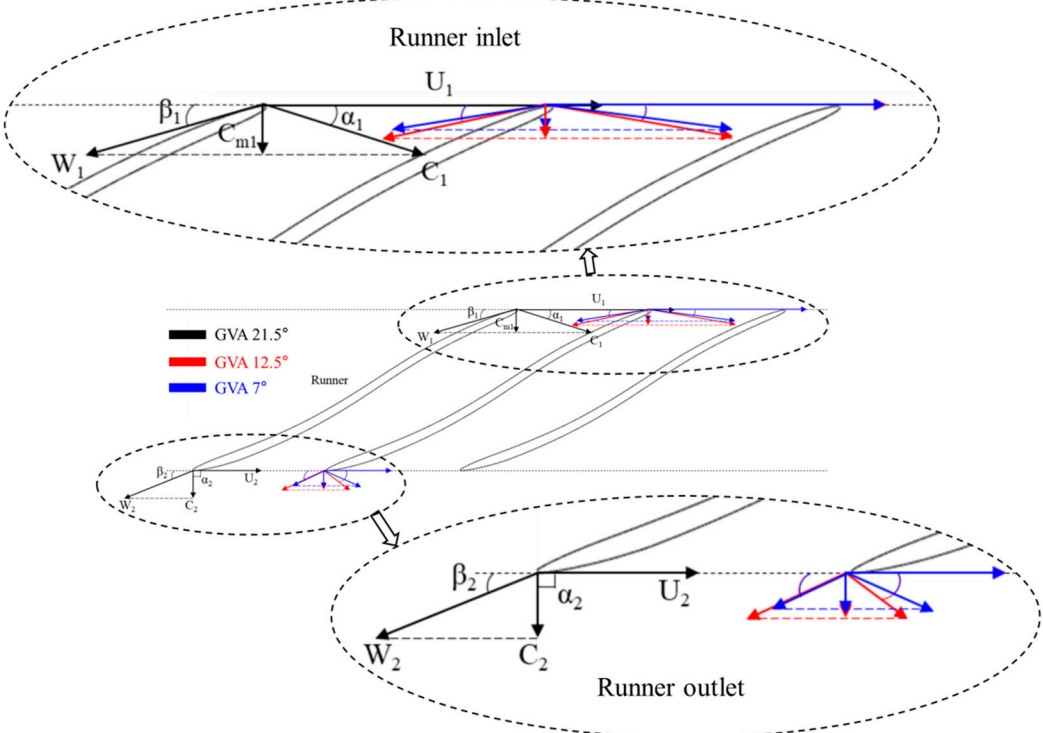

**Figure 8.** Velocity triangles at runner inlet and outlet with GVAs of 21.5°, 12.5°, and 7°.

In order to indicate the characteristics of the flow incidence angle according to the flow rate, Figure 9 shows the beta angle distributions at the runner inlet according to the GVAs as the different flow rate conditions. The beta angles were varied with each GVA as the flow rate changed. Therefore, the flow characteristics were influenced by changing the incidence angle between the blade and flow angle according to the GVAs as compared in Figure 8.

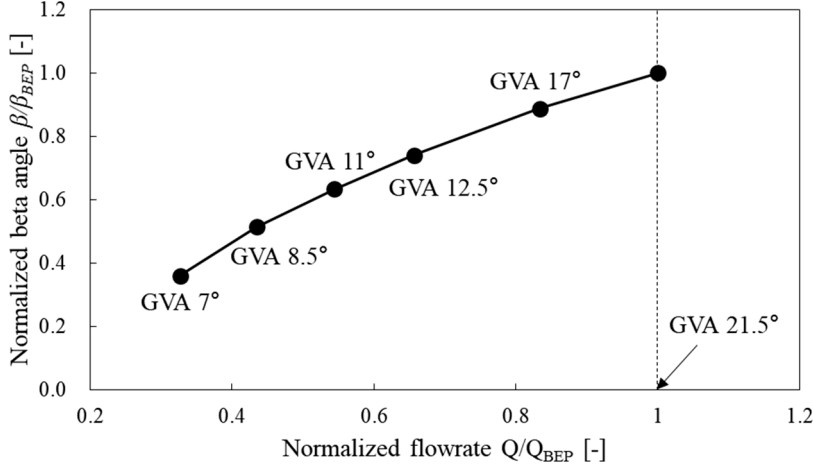

**Figure 9.** Beta angle distributions at runner inlet with GVAs.

In order to investigate the inter-blade vortex characteristics of the runner passages in turbine mode under low flow rate conditions, the velocity invariant Q distributions [23], which defines the local balance between the shear strain rate and the vorticity magnitude, were compared, as shown in Figure 10. The Q distributions at the hub and at 25% and 50% of the spans were investigated for each flow rate with reference to the flow stagnation regions that were largely contained within this region. Inter-blade vortices have been reported to attach to the intersection of the runner's leading edge with

the hub or mid-way from the hub between the blades close to the suction side [24,25]. In this analysis, as the flow rate decreased, the inter-blade vortices developed gradually from near the trailing edge to the leading edge, and the vortices were observed more clearly at the hub end of the span. In particular, distinct inter-blade vortices were found near the middle and the leading edge of the runner blade in the range of flow rates from 0.66 QBEP to 0.30 QBEP with GVAs of 12.5° and 7°, respectively. Thus, the inter-blade vortices developed in the passages of runner under low flow rate conditions, and these vortex characteristics can be confirmed by the difference in the incidence angle according to the flow rate shown in Figures 8 and 9.

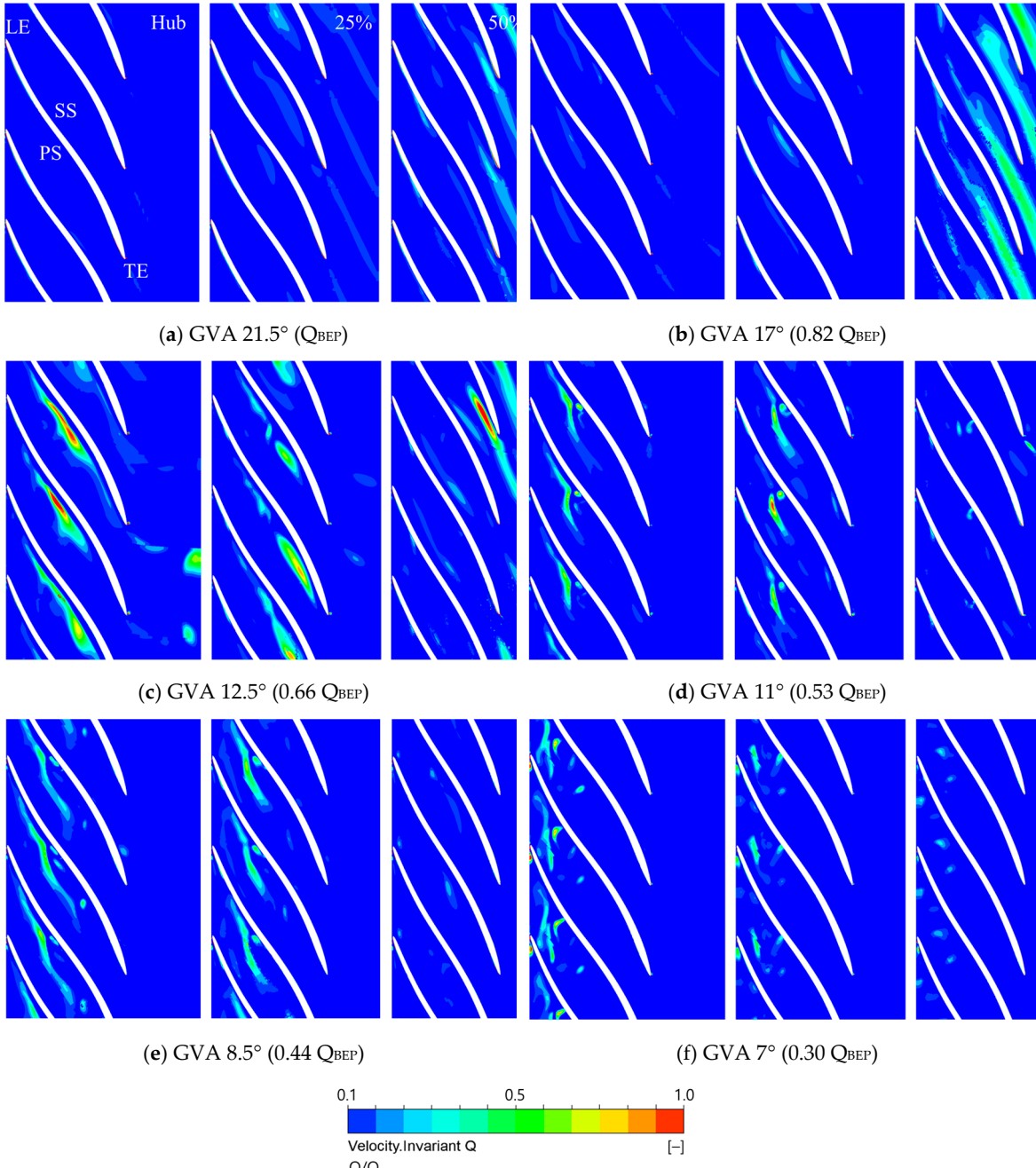

**Figure 10.** Invariant Q distributions at hub (left) and at 25% (middle) and 50% (right) of the spans, with GVAs of (**a**) 21.5°, (**b**) 17°, (**c**) 12.5°, (**d**) 11°, (**e**) 8.5°, and (**f**) 7°.

### 4.3. Vortex Rope Characteristics According to Flow Rate

The internal flow and vortex rope characteristics inside the draft tube under the low flow rate conditions depend on the flow and velocity characteristics at the outlet of the runner, which is closely related to the swirl intensity. Therefore, this study used the swirl number $S$, defined below, to indicate the ratio between the axial fluxes of angular and axial momentum as a representation of the swirl intensity characteristics at the runner outlet, as follows [26–29]:

$$S = \frac{\int_0^R C_m \cdot C_u \cdot r^2 \cdot dr}{R \int_0^R C_m^2 \cdot r \cdot dr} \tag{7}$$

where $C_m$ and $C_u$ are the time-averaged axial and tangential velocity, respectively; and $R$ is the radius of the reference section. Furthermore, the above swirl number S can be also calculated as a function of the axial flow velocity $C_m$ or the discharge $Q$, as follows [26–29]:

$$S = \frac{\omega R}{2} \times \left( \frac{1}{C_m} - \frac{1}{C_m^0} \right) \tag{8}$$

$$S = n_{ED} \frac{\pi^2}{8} \cdot \left( \frac{1}{Q_{ED}} - \frac{1}{Q_{ED}^0} \right) \tag{9}$$

$$Q_{ED} = \frac{Q}{D^2 E^{0.5}} \tag{10}$$

where, $C_m$ and $C_m^0$ are the axial flow velocity for observed and swirl-free conditions, respectively. $Q_{ED}$ and $Q_{ED}^0$ are the discharge factors from Equation (10) for observed and swirl-free conditions, respectively. The theoretical swirl-free condition represents the case in which the flow at the runner outlet is axial, and in this study, the BEP condition was considered as the swirl-free condition. Figure 11 shows the comparisons of the analytical and actual swirl number distributions for each flow rate condition using Equations (7) and (9), respectively and comparisons of the swirl number distributions using Equation (8) on the observed line in the draft tube as shown in Figure 4. As can be seen in Figure 11a, the both analytical and actual swirl number distributions increased maximally to 3.64 and 1.59, respectively, at a GVA 7°. In addition, the actual swirl numbers show the similar tendency with the analytical numbers when GVA is higher than 12.5°. However, when the GVA is lower than 12.5°, the actual swirl numbers show different tendency with the analytical swirl numbers. The analytical swirl numbers are tending toward $+\infty$ when the flow rate is becoming zero, which is lack of physical phenomenon. The difference between the analytical and actual results can be seen by the flow separation in the runner passages with changed relative flow angle at the runner outlet [29]. Previously, Favrel et al. [29] observed similar swirl number distributions between the analytical and experimental results. In the Figure 11b, the local swirl number distributions are shown with various GVAs on the observed line (marked in Figure 4) from center to the wall in the draft tube. There are large discrepancies between the swirl number distributions in Figure 11a and on the lines in Figure 11b. It can be seen that the analytical and actual swirl numbers were averaged by the passage area and then calculated, while the swirl numbers on the line were calculated by the local values. Therefore the local results can be shown with a lager values. The difference between the maximum and minimum values of the swirl number was increased gradually and their differences were moved from center to wall of the draft tube, as the GVAs decreased. It can be confirmed with the absolute flow angle at the outlet of runner as shown in Figure 8; as the absolute flow angle decreased (as the GVA decreased), higher swirl characteristics also were moved toward the draft tube wall.

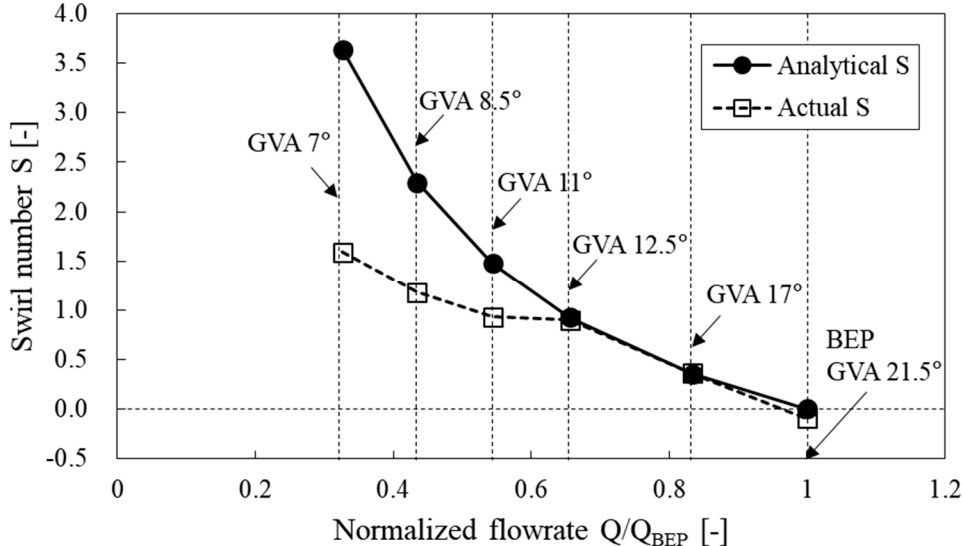

(**a**) Analytical and actual swirl number distributions

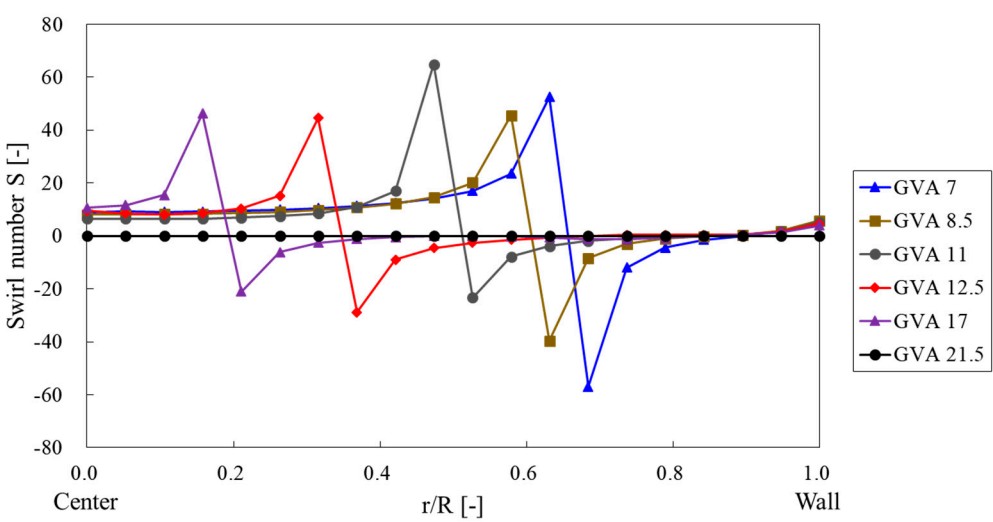

(**b**) Local swirl number distributions on the observed line

**Figure 11.** Calculated swirl numbers from (**a**) analytical and actual; and on the (**b**) observed line for various GVAs.

Figure 12 shows the iso-surface distributions of the pressure in the draft tube from the unsteady-state analyses. The value of pressure was decided with the relative water saturation pressure by considering the water level of lower reservoir. The developed vortex rope shapes were compared qualitatively according to the flow rates with various GVAs. The largest vortex rope was formed at the GVA of 12.5°, with a swirl number of 0.89. The size of the vortex rope decreased gradually as the flow rate decreased until the GVA of 8.5°, with a swirl number of 1.17, and finally the vortex rope disappeared visibly at the GVA of 7°, with a swirl number of 1.59. Although the swirl intensity increased as the flow rate decreased, distinct vortex ropes were shown clearly in the flow rate range of $0.66\,Q_{BEP}$–$0.44\,Q_{BEP}$; despite the stronger swirl intensity under very low flow rate conditions, thevortex rope disappeared visibly at $0.30\,Q_{BEP}$.

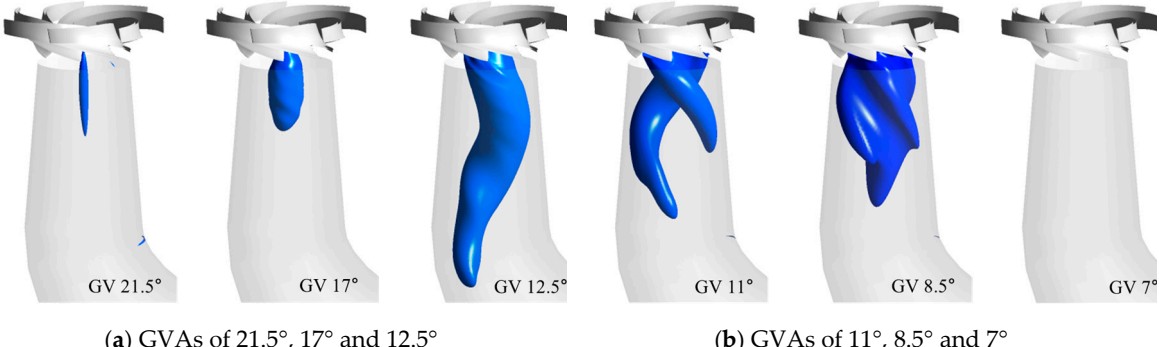

(**a**) GVAs of 21.5°, 17° and 12.5°  (**b**) GVAs of 11°, 8.5° and 7°

**Figure 12.** Iso-surface distributions in the draft tube with GVAs of (**a**) 21.5°, 17°, and 12.5°, (**b**) 11°, 8.5°, and 7°.

To confirm the cause of the flow phenomena in the draft tube according to the flow rates, the velocity triangle components were compared at the outlet of runner as shown in Figure 8. The absolute velocity component at the runner outlet flowed in the axial direction with $\alpha_2$ as a right angle at the GVA of 21.5°. In contrast, the absolute velocity components at the GVAs of 12.5° and 7° flowed not only in the axial direction but also in the radial direction at the outlet of runner because the $\alpha_2$ angle decreased as the flow rate decreased. The increase in the radial velocity component at the outlet of runner can be considered as the cause of the increase in the swirl intensity under low flow rate conditions.

Figure 13 shows the streamline distributions of the time-averaged velocity values from the unsteady state-analyses in the BEP condition, the 0.66 $Q_{BEP}$ condition with the larger vortex rope, and the 0.30 $Q_{BEP}$ condition with the highest swirl number. In the Figure 13a, the internal flow in the BEP condition flowed in the axial direction, like the absolute velocity component at the runner outlet, as in the velocity triangle distributions shown in Figure 8. In Figure 13b, the complicated internal flow characteristics were shown in the draft tube due to the relatively increased swirl intensity, including the development of the large vortex rope with precession indicated by the dotted line. However, in Figure 13c, the 0.30 $Q_{BEP}$ condition with the highest swirl number does not show the vortex rope characteristics with precession because the swirl intensity was too strong and the internal flow became very complicated in the draft tube. Therefore, the vortex rope characteristics in the draft tube did not show visibly in proportion to the swirl intensity at the outlet of runner; however, the results confirmed the visible development of a vortex rope within a certain range of swirl intensities and flow rates.

Figure 14 shows the quantitative distributions of absolute and relative flow angles along the spanwise (hub to shroud) direction at the runner outlet. The angles were normalized by each maximum flow angle. In the Figure 14a, the normalized absolute flow angle with GVA of 21.5° shows the almost zero; and the absolute flow angles were increased as the flow rate decreased as shown in Figure 8 (velocity triangle) and Figure 13 (3D streamline). Meanwhile, the relative flow angles were made by along the angle of runner blade TE; however, because of the complicated flow at the low flow rate condition, the normalized relative flow angles show different distributions as shown in Figure 14b. It can be confirmed that discrepancy between the actual and analytical swirl number at low flow rate region as shown in Figure 11a.

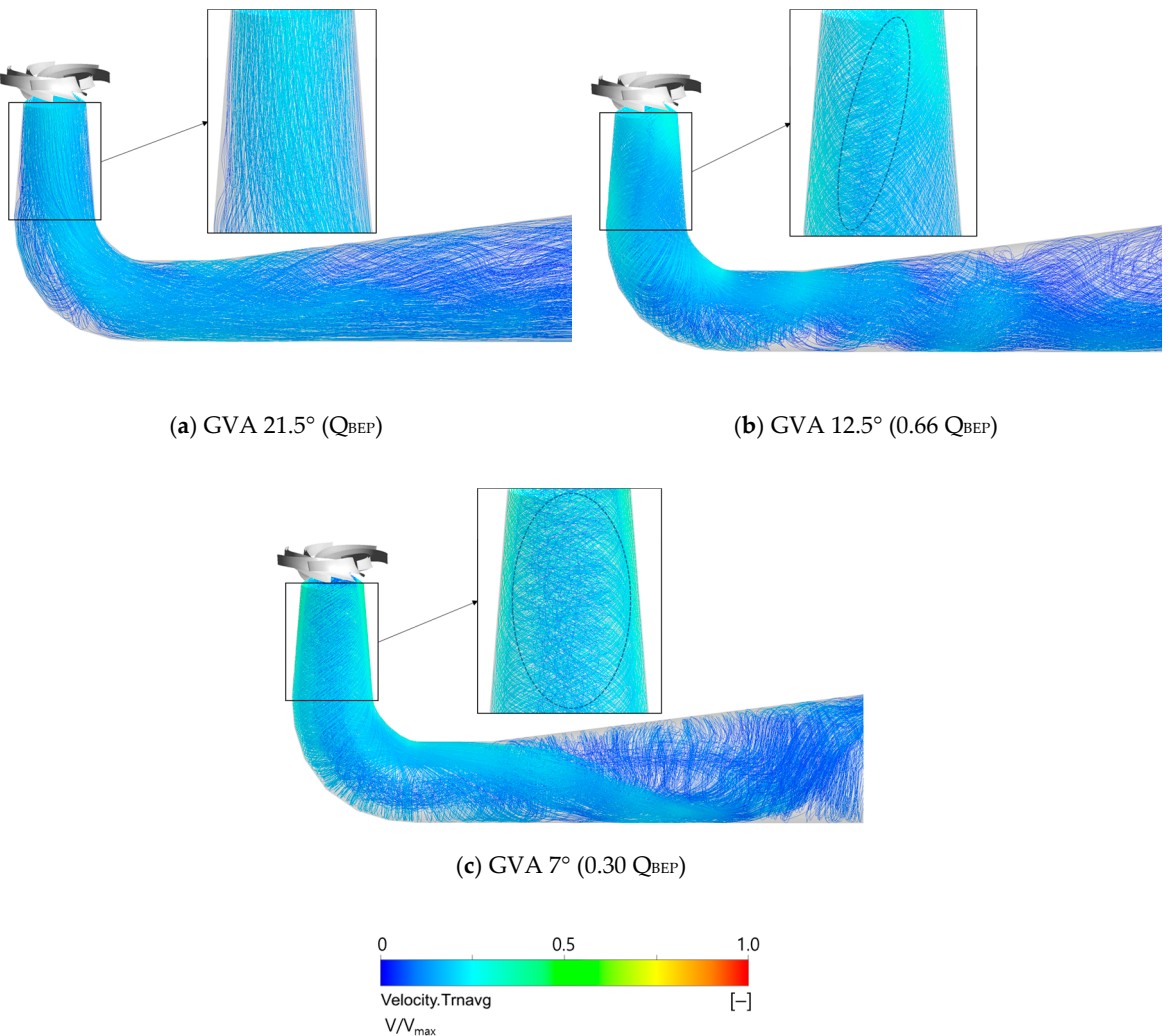

**Figure 13.** 3D streamline distributions in the draft tube with GVAs of (**a**) 21.5°, (**b**) 12.5°, and (**c**) 7°.

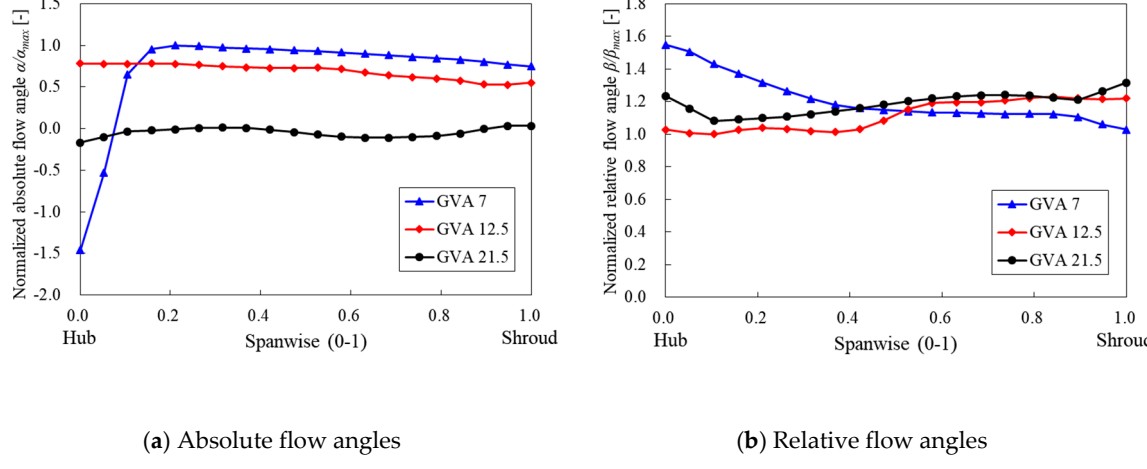

(**a**) Absolute flow angles          (**b**) Relative flow angles

**Figure 14.** Distributions of (**a**) absolute and (**b**) relative flow angles along the spanwise direction at the runner outlet with GVAs of 21.5°, 12.5°, and 7°.

Figure 15 compares the time-averaged axial, circumferential velocity and static pressure distributions along the line from the draft tube's center point to the wall at the height of the measuring point p3 shown in Figure 4. The abscissa represents the measurement location relative to the radius

from the center (0) to the wall (1) of the draft tube, and the values of velocities and pressures were normalized by the maximum value. The GVA of 21.5° shows a relatively even axial velocity distribution for the BEP condition, whereas backflows are shown near the center of the draft tube with GVAs of 12.5° and 7° as shown in Figure 15a. In particular, the backflow showed a wider range from the center to approximately r/R = 0.65 with the GVA of 7°, which had the highest swirl intensity, relative to the GVA of 12.5°, which showed the large vortex rope. In Figure 15b, the circumferential velocity distributions appear relatively lower, at close to zero, with the GVAs of 21.5° and 7° that did not show the vortex rope, whereas a large circumferential velocity appears over a wider range, with the GVA of 12.5°. In addition, in the Figure 15c, the pressure with GVA of 12.5° shows relative lower at near center, which location showed the development of the vortex rope. These results show the influence of revolution with precession on the development of the vortex rope. Thus, the axial velocity results show that the backflow that adversely affected the smooth flow and performance in the draft tube occurred, not only in the specific flow rate range in which the vortex rope developed but also in the low flow rate region with the strongest swirl intensity and very complicated flow.

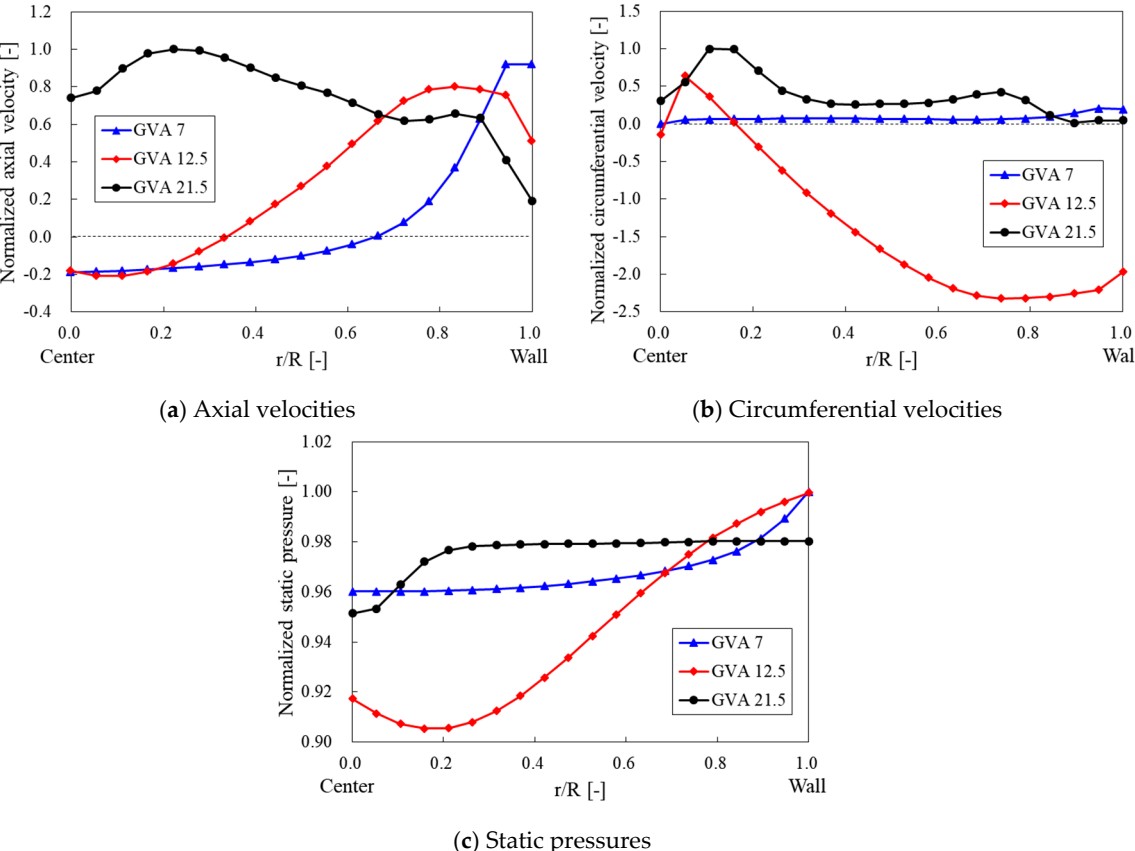

(**a**) Axial velocities

(**b**) Circumferential velocities

(**c**) Static pressures

**Figure 15.** Distributions of (**a**) axial, (**b**) circumferential velocities, and (**c**) static pressures on observed line with GVAs of 21.5°, 12.5°, and 7° in the draft tube.

### 4.4. Correlation Between Inter-Blade Vortex and Vortex Rope

Figure 16 shows the internal flow characteristics through the meridional velocity and velocity streamline distributions on the hub span during the last revolution of the runner in 90° intervals. These results were taken from the unsteady-state analyses showing stable periodicity for GVAs of 12.5° and 7° (0.66$Q_{BEP}$ and 0.30$Q_{BEP}$, respectively). The meridional velocity was normalized by the local maximum meridional velocity (40 m/s) of the runner domain. Differences in the meridional velocity were shown to be generally due to the differences in flow rates for each GVA. The flow stagnation regions and blockage effects in the 0.30 QBEP condition were shown to be due to the developed inter-blade vortices

at the near leading edge, which is not flow properly through the passages in the streamline distributions. In the 0.66 QBEP condition, the internal flow was not smooth because blockage effects due to the inter-blade vortices occurred in the middle of the runner passages. In addition, inter-blade vortices developed continuously at the same location with lower velocity regions, with a slight change in vortex size according to the flow rate; these vortices affected the flow in the passages and outlet of the runner. With the GVA of 7°, the inter-blade vortices near the leading edge had a larger effect on the flow in only the runner passages, whereas with the GVA of 12.5°, the inter-blade vortices in the middle of the runner passages had a relatively large influence on the internal flow and flow rate in both the runner passage and outlet, thereby changing the flows at the runner outlet during the runner's revolution. These results show that the inter-blade vortices develop at different locations in the runner passages according to the flow rate, and continue to develop at the same location over time although changing slightly in size, inducing blockage effects and complicating internal flow in the runner passages and outlet.

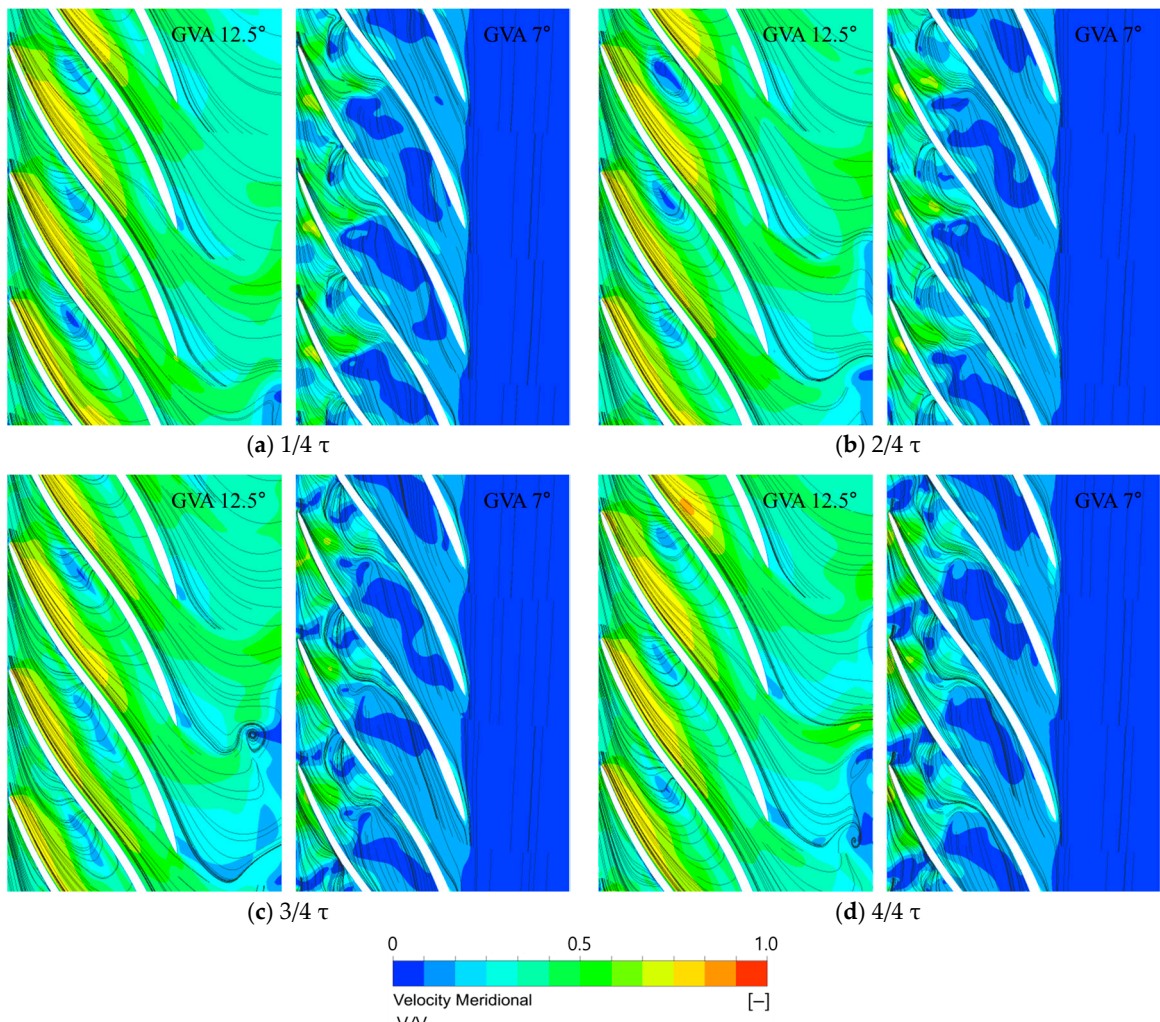

**Figure 16.** Meridional velocity distributions on the hub span of the runner with GVAs of 12.5° (**left**) and 7° (**right**) during one revolution of the runner: (**a**) 1/4 τ, (**b**) 2/4 τ, (**c**) 3/4 τ, and (**d**) 4/4 τ.

In order to observe the flow structure characteristics of the fully developed vortex rope during the last revolution of the runner under each flow rate condition, the vortex ropes were compared at the GVAs of 17°, 12.5°, and 11° as shown in the Figure 17. The iso-surface distributions of the pressure as the flow structure were decided with the relative water saturation pressure by considering the water level of lower reservoir. The fully developed vortex rope shapes in each flow rate condition were very

similar in length and size, and the vortices maintained these shapes and rotated in the same rotational direction of the runner in the draft tube.

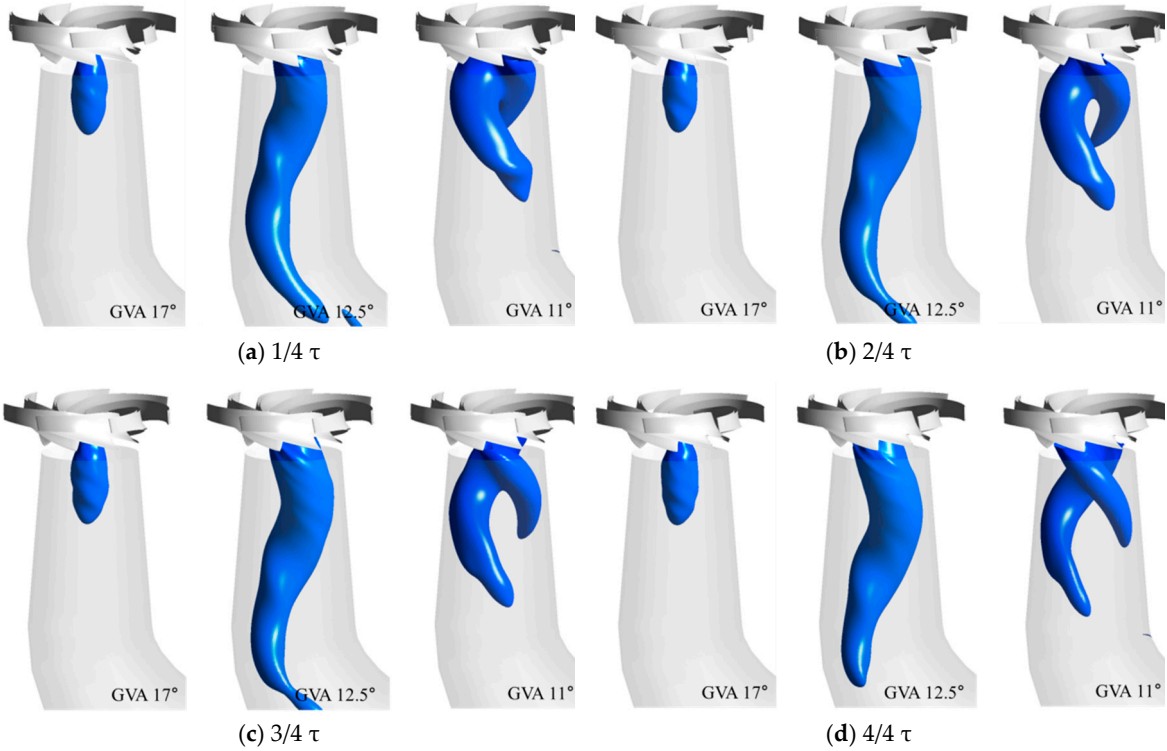

**Figure 17.** Iso-surface distributions of vortex rope with GVAs of 17° (**left**), 12.5° (**middle**), and 11° (**right**) during one runner revolution: (**a**) 1/4 τ, (**b**) 2/4 τ, (**c**) 3/4 τ, and (**d**) 4/4 τ in the draft tube.

This study confirmed that the inter-blade vortex characteristics in the runner passages depend on the flow rate conditions and velocity components at the runner inlet, and the vortex rope characteristics in the draft tube occur in specific flow rate ranges and velocity conditions at the outlet of runner. The flow and velocity characteristics at the outlet of runner are influenced from the internal flow characteristics in the runner passages as shown in Figures 11 and 14, and to investigate the correlation of the inter-blade vortex and vortex rope, Figure 18 compares the iso-surface and 3D streamline flow distributions from the runner passages to the draft tube during the one runner revolution with GVAs of 12.5° and 7°. Here, the iso-surface is the flow stagnation region shown in Figure 6. With the GVA of 12.5°, the flow stagnation regions occurred in the runner passages; irregular flow and flow stagnation regions occurred at the runner outlet; and these flows were connected with the top of vortex rope in the draft tube. With the GVA of 7°, the flow stagnation regions developed in the runner passages, with complicated flow; however, the vortex rope was not developed visibly. Although the swirl intensity increased as the flow rate decreased at the runner outlet, when an inter-blade vortex developed in the middle of the runner passage, within the specific range of low flow rates (0.82 $Q_{BEP}$–0.44 $Q_{BEP}$), the internal flow in the runner passage affected the flow angle distribution and the swirl characteristics at the runner outlet and in the draft tube, as shown in Figures 11 and 14, resulting in the visible development of the vortex rope.

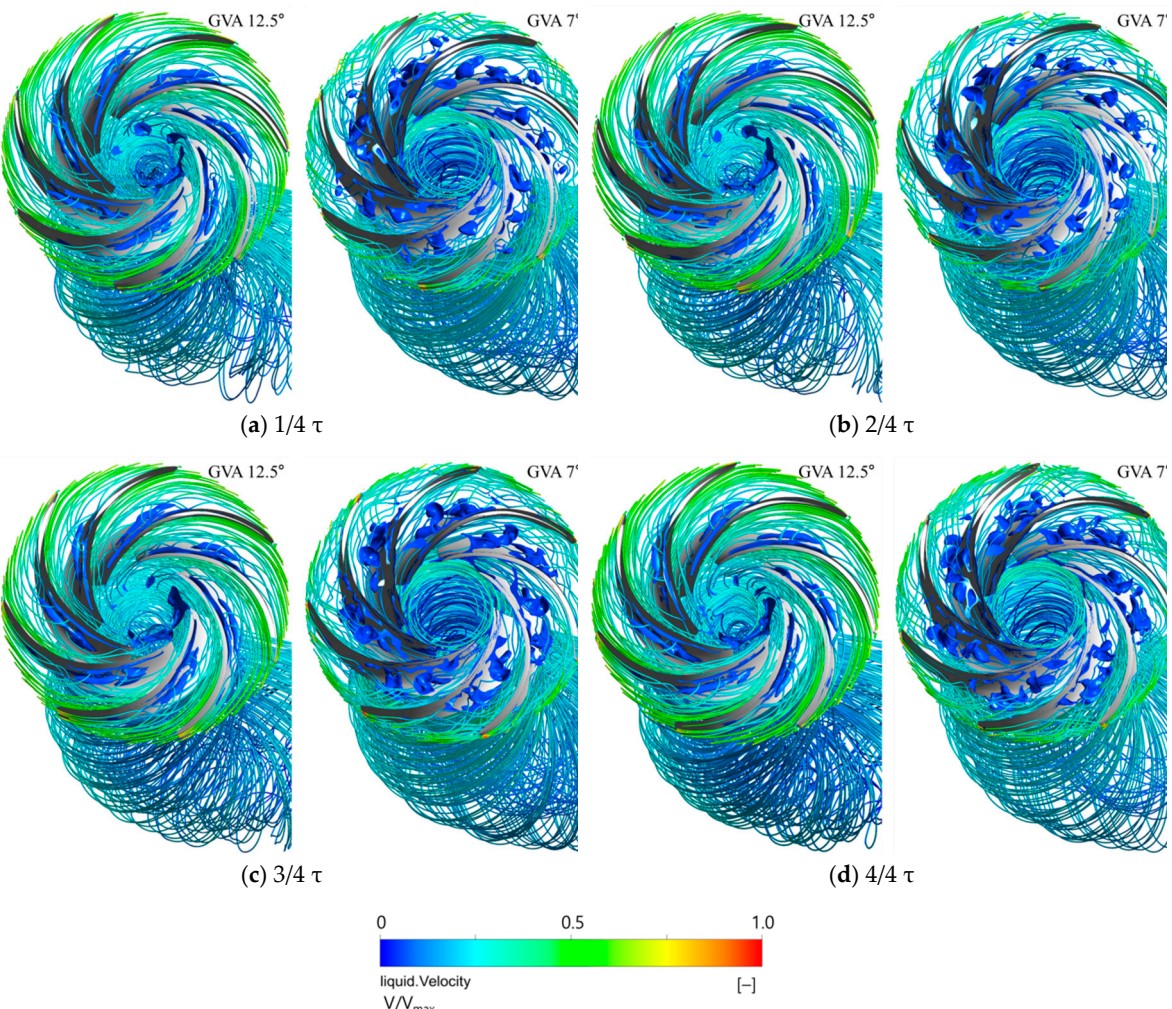

**Figure 18.** Iso-surface and 3D streamline flow distributions in the runner passages and draft tube with GVAs of 12.5° (**left**) and 7° (**right**) during one runner revolution: (**a**) 1/4 τ, (**b**) 2/4 τ, (**c**) 3/4 τ, and (**d**) 4/4 τ.

### 4.5. Unsteady Pressure Characteristics of the Inter-Blade Vortex and Vortex Rope

Figure 19 shows the unsteady pressure characteristics obtained via fast Fourier transformation (FFT) analysis with GVAs of 21.5°, 12.5°, and 7° at measuring points 01, 07, and 14 on the outlet of guide vane outlet, indicated in Figure 4. The pressure fluctuation was normalized by the Equation (11) to compare all operating conditions, and the frequency was normalized by the rotational frequency ($f_n$) of the pump-turbine.

$$C_p = \frac{p}{\rho E} \tag{11}$$

with the GVA of 7°, which exhibited the inter-blade vortex at the near leading edge, a relatively higher pressure characteristic was shown at the first blade passing frequency (BPF) and in the low-frequency region, normalized to $3.8f_n$, relative to the other observed GVA conditions. However, with the GVAs of 12.5° and 7°, relatively low first BPF values were shown at measuring point 14, which was a different tendency compared to the other measuring points, although with the GVA of 7°, the unsteady pressure distribution in the low-frequency region, normalized to $3.2\,f_n$, showed characteristics similar to those at other measuring points. Differences among the measuring points, such as the unsteady pressure at the normalized frequencies of $3.2\,f_n$ to $3.8\,f_n$, were attributed to changes in the inter-blade vortices that complicated flow near the runner's leading edge during the revolution, as shown in Figure 16.

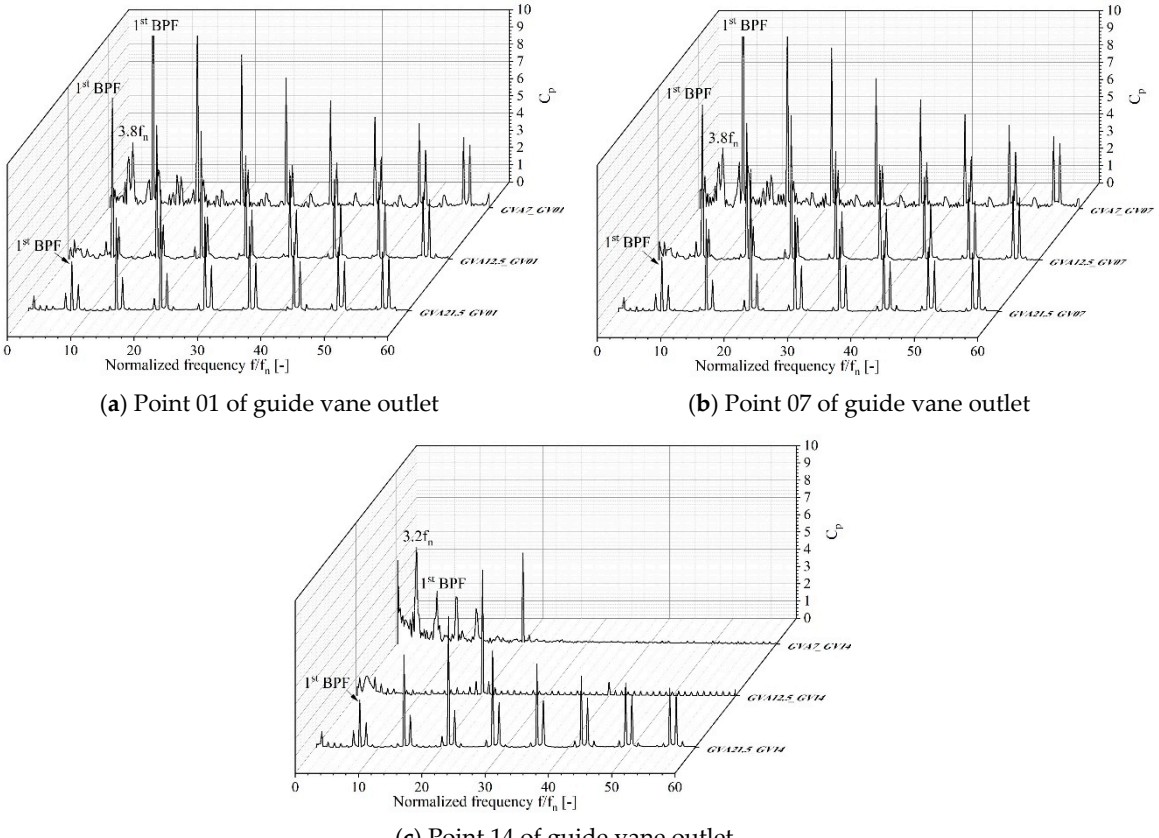

(**a**) Point 01 of guide vane outlet　　　　　　　　(**b**) Point 07 of guide vane outlet

(**c**) Point 14 of guide vane outlet

**Figure 19.** Normalized unsteady pressure distributions with GVAs of 7°, 12.5°, and 21.5° at the guide vane outlets monitored with measuring points (**a**) 01, (**b**) 07, and (**c**) 14.

Figure 20 compares the unsteady pressure characteristics with GVAs of 12.5° and 7° at the measuring points L1 through L4 from point 2 on the draft tube cone, where the pressure fluctuation and frequency were normalized using the same methods as for the data shown in Figure 19. The unsteady pressure characteristics of the draft tube are generally approximately five times lower than at the guide vane outlet shown in Figure 19 because the guide vane outlet is a higher-pressure section before the runner whereas the draft tube is a lower-pressure section after the runner. In Figure 20a, similar unsteady pressure characteristics are shown at all four measuring points with the GVA of 7°, under which condition the vortex rope was not generated visibly in the draft tube, but a relatively higher pressure characteristic was shown in the low-frequency region, normalized to $0.2\ f_n$ and $3.2\ f_n$ prior to the first BPF, due to the strong swirl intensity. Here, the normalized frequency of $3.2\ f_n$ can be attributed to unsteady pressure characteristics induced by the inter-blade vortex in the runner passages as shown in Figure 16. Figure 20b shows similar unsteady pressure characteristics at the four measuring points, although small differences appear because of the development and revolution of the vortex rope, where higher pressure characteristics occurred in the lower-frequency region relative to the GVA of 7°. This result can be interpreted to indicate that unsteady pressure characteristics were induced by the development and revolution of the vortex rope. Liao et al. [12] and Kirschner et al. [14] observed similar unsteady pressure characteristics in the low-frequency region due to a vortex rope by numerical analysis.

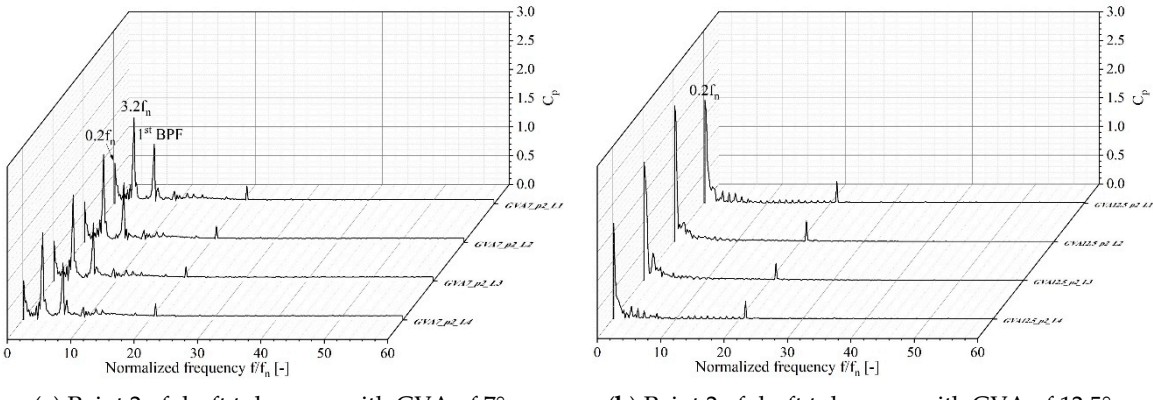

(**a**) Point 2 of draft tube cone with GVA of 7°    (**b**) Point 2 of draft tube cone with GVA of 12.5°

**Figure 20.** Normalized unsteady pressure distributions at the measuring points of L1, L2, L3, and L4 of the draft tube at point p2 with GVAs of (**a**) 7° and (**b**) 12.5°.

In order to indicate the unsteady pressure characteristics along the flow direction in the draft tube, Figure 21 compares the unsteady pressures that occurred with GVAs of 21.5°, 12.5°, and 7°, at the measuring points of p1 through p4 at the location L1 indicated in Figure 4. The pressure fluctuation and frequency values were normalized using the same methods as for the data shown in Figure 19. In Figure 21a, a high pressure characteristic is shown at the first BEP with the GVA of 21.5°, whereas relatively high pressure characteristics in the low-frequency region of 0.4 $f_n$ and 3.2 $f_n$ are shown with the GVAs of 12.5° and 7°, respectively. In particular, the relatively dominant pressure characteristic was shown in the lowest-frequency region of 0.2 to 0.4 $f_n$ with the GVA of 12.5°, which had the largest vortex rope. In Figure 21b–d, as the measuring point moved in the flow direction, the unsteady pressure characteristics decreased gradually, showing similar trends to the GVAs of 21.5° and 7°. However, with the GVA of 12.5° that had the largest vortex rope, the pressure characteristic decreased from point p4 after increasing until point p3. This result occurred because as the vortex rope developed and rotated, it became larger closest to point p3, and the unsteady pressures decreased as the size of vortex rope decreased and moved away from the draft tube wall along the flow direction. Therefore, despite the highest swirl intensity shown with the GVA of 7°, the relatively dominant unsteady pressure characteristics were shown in the low-frequency region induced by the largest vortex rope with the GVA of 12.5°.

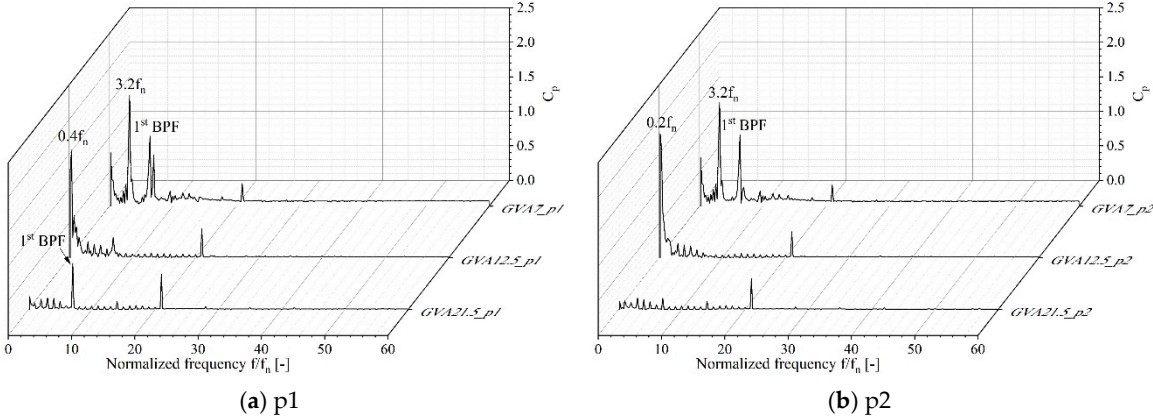

(**a**) p1    (**b**) p2

**Figure 21.** *Cont.*

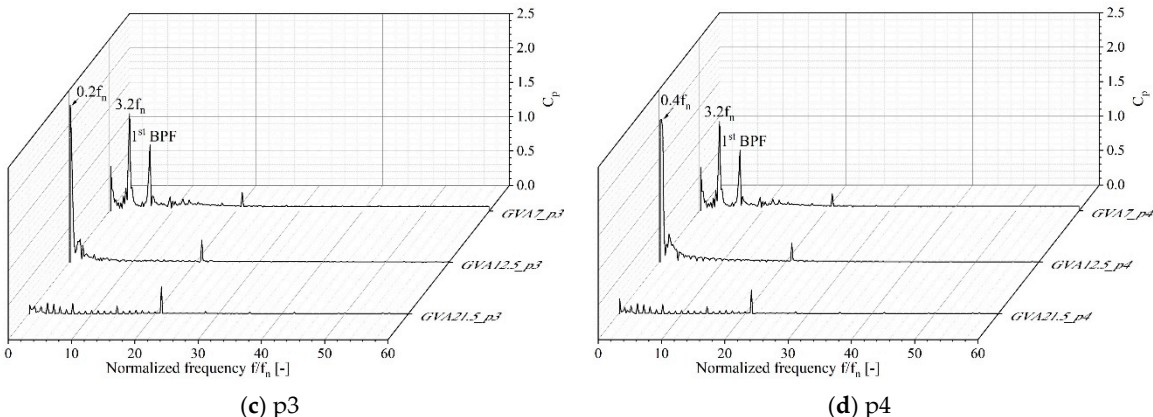

**Figure 21.** Normalized unsteady pressure along the flow direction in the draft tube with GVAs of 21.5°, 12.5°, and 7° at measuring points (**a**) p1, (**b**) p2, (**c**) p3, and (**d**) p4.

## 5. Conclusions

This study investigated the internal flow and unsteady pressure characteristics of the inter-blade vortex and vortex rope in turbine mode of a pump-turbine via steady- and unsteady-state 3D RANS analyses conducted under low flow rate conditions. In addition, the inter-blade vortex and vortex rope characteristics were compared with the internal flow and velocity characteristics to observe the cause of their correlations.

The inter-blade vortices developed in the runner passages as the flow rate decreased, and changing velocity components at the runner inlet were confirmed according to the flow rates through velocity triangle distributions. The inter-blade vortices developed at different locations in the runner passages depending on the flow rates, and continued to occur at those locations over time, with slight changes in size. In addition, the internal flow in the runner passage affected the flow angle distribution at the runner outlet and the swirl characteristics in the draft tube.

The flow structure and development of the vortex rope were confirmed according to the flow rate in the draft tube by applying the swirl number definition to represent the swirl intensity. In addition, the cause of swirl characteristics was confirmed through velocity triangle and flow angle distributions indicating the flow characteristics at the runner outlet under low flow rate conditions. This analysis showed that the vortex rope with precession did not develop in the draft tube under the highest swirl intensity conditions at the lower flow rates because the highest swirl intensity complicated the flow where the vortex rope would have developed. However, although the vortex rope did not develop visibly with the highest swirl intensity, a relatively wide range of back flow was indicated in the draft tube. The development of the inter-blade vortices and vortex rope depended on the flow and velocity characteristics of each inlet and outlet of runner, and within the specific flow rate range where the inter-blade vortex developed in the middle of the runner passages, the flow in the runner passages affected the flow characteristics at the runner outlet and led to the visible development of the vortex rope.

The unsteady pressure characteristics at the runner and draft tube were confirmed through FFT analysis. High unsteady pressures were observed in relatively low-frequency regions under low flow rate conditions. In particular, under the conditions that led to the largest vortex rope, a dominant unsteady pressure was shown in the relatively lower-frequency region of 0.2 to 0.4 $f_n$ in the draft tube, and these unsteady pressure characteristics were relatively higher than when the vortex did not develop.

From these results, the turbine mode operation of a pump-turbine can be expected to exhibit a vortex rope with precession in the flow rate range of 0.44 $Q_{BEP}$–0.82 $Q_{BEP}$, as inter-blade vortices develop in the middle of the runner passages. These vortices can adversely affect the system's operation



because of the high unsteady pressure, and therefore the vortex characteristics must be considered to ensure stable operations under off-design low flow rate conditions.

**Author Contributions:** Conceptualization, validation, investigation, data curation, S.-J.K.; J.-W.S.; Y.-S.C.; J.P.; N.-H.P. and J.-H.K.; resources, Y.-S.C.; J.P.; N.-H.P. and J.-H.K.; writing—original draft preparation, S.-J.K.; J.-W.S. and J.-H.K.; writing—review and editing, supervision, project administration, J.-H.K.; funding acquisition, J.P.

**Funding:** This research was funded by the Korea Hydro & Nuclear Power Co. Ltd. (grant number L17S029000).

**Conflicts of Interest:** The authors declare no conflict of interest.

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
