# Peer review of "Inter-Blade Vortex and Vortex Rope Characteristics of a Pump-Turbine in Turbine Mode under Low Flow Rate Conditions"

_water, doi:10.3390/w11122554_

Round 1
Reviewer 1 Report
The review of the article can be found in the following file.

Reviewer 2 Report
The reversible hydraulic machines are extensively investigated due to their abilities to answer to the energy market requirements operating both in pumping-turbine modes. The operation of the pump-turbine machines on wide operation range is required by the energy market. However, the operation of the pump-turbine machines at low flow rate conditions are hindered by unsteady phenomena (e.g. vortex rope, inter-blade vortices).
The main purpose for the actual investigations is defined in the last sentence of the first section. “The flow structures, correlations, and unsteady pressure characteristics of the inter-blade vortex and vortex rope were investigated according to flow rate, and the swirl number at the outlet of runner”. Several qualitative pictures are provided for different guide vane openings. However, the quantity (quantities) associated to the flow structures should be defined then their evolution should be assessed against the flow rate and/or the swirl number, respectively. The manuscript is followed with difficulty by the reader due to a lot of pictures are provided without relevant data. Several data about numerical investigations included in the manuscript are unclearly presented/explained. As a result, several recommendations are proposed to improve the scientific level and the clarity of the manuscript. A MAJOR REVISION is recommended based on the analysis performed.
The main parameters associated to the pump-turbine investigated in this case are given in Table 1 according to the IEC 60193 standard.
It is mentioned that the three-dimensional (3D) steady- and unsteady-state RANS numerical investigations were performed using a k–ω-based SST turbulence model with two-phase flow (cavitating two-phase flow) using Rayleigh-Plesset equation.
R1: The equations solved and the cavitating two-phase flow model (e.g. Singhal et al., Zwart-Gerber-Belamri, Schnerr and Sauer) selected in this investigation should be presented together with the boundary conditions imposed for both phases (liquid and gaseous). All parameters corresponding to their numerical setup should be presented (e.g. vaporization and condensation constants for cavitating model, bubble radius, nucleation site volume fraction, the relaxation factor for vapor/gaseous, the effect of non condensable gases was taken into account and so on). The authors should detail the boundary conditions imposed in numerical simulations to be correlated with the experimental investigations (e.g. boundary conditions on the inlet/outlet, reference static pressure selected in the numerical simulation to be correlated with the cavitation number on the test rig and so on). The reader can learn from the authors’ experience with the numerical setup for cavitating flows increasing the chance to be referred their work. A meridian section view should be added in Figure 1 to evidence the main geometrical parameters of the hydraulic passage of the pump-turbine investigated in this case.
The mesh refinement study is performed (see Figure 3) while the global parameters (normalized power and normalized efficiency) determined based on numerical simulation are compared against experimental data (see Figure 5).
R2: The equations used to determine both power and efficiency quantities presented in Figure 5 should be given in the paper. The authors should explain what type of efficiency (e.g. hydraulic or turbine) is yielded based on the numerical simulation as well as from experimental investigation. The uncertainty bars should be added on experimental data. Also, the mesh distribution on each part (e.g. spiral casing, stay vane and guide vane regions, runner and draft tube) of the computational domain should be given in a table.
Several measuring points (at the outlet of guide vane and on the cone of draft tube) have been selected to investigate the unsteady pressure field under various low flow rate conditions, see Figure 4.
R3: The positions of the pressure taps indicated in Figure 4 should be marked on the meridian view of the pump-turbine together with the dimensionless length for each tap. The dimensionless length should be computed using the reference diameter of the runner (D2e) according to the IEC 60193. The dimensionless length for each pressure tap located on the cone wall should be referred with respect to the runner outlet while one single dimensionless value should be indicated on this meridian view for all three pressure taps (e.g. GV_01, GV_07, GV_14) located between the guide vane blades and the leading edge of the runner. Please indicate the dimensionless length of the cone, too. The dimensionless values are supporting the readers to compare with other data referring to this paper.
The streamlines associated to the absolute flow field are plotted in the guide vanes while the streamlines associated to the relative flow field are detailed in the runner as well as the iso-surfaces are determined (see Figure 6) based on the time-averaged values of the velocities from the unsteady-state analyses. The figures presented in Figure 6 for several discharge values give a general view about the positions of the stagnant regions developed in the runner.
R4: The authors should mention the value selected for the stagnation regions plotted in Figure 6. The volume of the stagnant region(s) should be determined for each discharge value in order to quantify its evolution. Is the static pressure low enough in vortex cores detected in the runner to develop the vapor/gaseous phase?
The velocity triangle for three discharge values were compared on the runner mid-span to explain the cause of the vortex development and flow stagnation regions in the runner channels, see Figure 7.
R5: It is appreciated the explanation given by authors to justify the development of the stagnant regions in the runner channels. However, this theoretical explanation with the velocity triangle located on or near to the leading edge of the blade is limited due to two reasons. (1) One reason is linked with the local flow near to the leading edge. The velocity triangle is valid for the flow located far enough to the blade. Therefore, the local flow near to the leading edge is different than the flow located far enough to the blade. The space between the trailing edge of the guide vane and the leading edge of the runner blade for pump-turbines is pretty small. Consequently, it is well known that strong rotor-stator interaction phenomena are associated to these hydraulic machines. (2) The second reason is associated to the leading edge geometry. The stagnant point distribution from the hub (crown) to the shroud (band) of the flow on the runner blade for one operating point can be plotted to check the design. A stagnant point distribution located on the leading edge of the runner blade at the best efficiency point (BEP) supports an appropriate design of the leading edge. For instance, an old selection of the leading edge geometry due to the technological reasons provides an improper distribution of the stagnant point on the Francis runner blade (on the suction side near to the crown and on the pressure side near to the band) at BEP, Muntean et al. (2010). As a result, the flow separation and hydrodynamic instabilities are developed on an extended operating domain with unsteady phenomena leading to a catastrophic event on the runner, Frunzaverde et al. (2010). The movement of the stagnant point on the pressure/suction side of the runner blade at part/full load conditions support flow separation and unsteady phenomena. Please revise this analysis of the hydrodynamic field in the runner to explain better the phenomena developed in the runner.
It is mentioned on page 8 that “the inter-blade vortices are developed gradually from the trailing edge to the leading edge as the flow rate is decreased. Also, the vortices were observed more clearly at the hub end of the span. In particular, distinct inter-blade vortices were found near the middle and the leading edge of the runner blade in the range of flow rates from 0.66QBEP to 0.30QBEP with GVAs of 12.5° and 7°, respectively. Thus, the inter-blade vortices developed in the passages of runner under low flow rate conditions”. In my opinion, the three-dimensional distribution of the inter-blade vortices cannot be properly understood by the reader looking to Q maps on three planes located at 0%, 25% and 50% to the hub (crown) as in Figure 8.
R6: A three-dimensional view of the hydrodynamic structure with iso-constant Q value on a runner channel might be more useful than Figure 8. Particularly, it is suggested to be considered a dimensional value of the Q instead of dimensionless one in order to show the difference between different operating points. It is expected that the same value of Q iso-surface to provide small or negligible volumes at operating points near to BEP and large volumes at part load conditions. In this way, the reader can perform a direct comparison between the shape and the volume of the inter-blade vortices for each dimensionless discharge value.
The swirl number is computed using a simplified equation form. This equation has been developed for experimental investigations where the quantities (n, Q and H) are obtained based on measurements.
R7: The swirl number should be computed based on numerical results obtained in the draft tube cone, see Figure 12. The hydrodynamic field is available for each operating point then the swirl number can be computed using both axial and tangential flux of moment of momentum, Resiga et al. (2006). As a result, the swirl number computed based on numerical data should be compared with data available in Figure 9. The tangential component, the absolute/relative flow angle and the static pressure radial distribution should be added in Figure 12.
The iso-surface distributions of the time-averaged values of the pressure in the draft tube from the unsteady-state analyses are shown in Figure 10. The vortex rope shapes were compared qualitatively according to the flow rates with various GVAs in Figure 14.
R8: The main purpose of the draft tube is to recover as much as possible of the static pressure available to the inlet with minimum hydraulic losses. Usually, the maxim static pressure is recovered along to the draft tube cone at best efficiency point (BEP). Moreover, the static pressure distribution is different from one operating point to another one. Therefore, the iso-pressure surface is faster closed for the operating points where the static pressure is better recovered that other operating point at partial load conditions. Also, the static pressure in the draft tube is directly correlated with the static pressure imposed on it computational domain. It is well known that the Navier-Stokes equation solved in the numerical simulation includes the pressure term where the pressure gradient is relevant quantity. Therefore, a reference static pressure has to be imposed on the computational domain. The imposed value of the reference static pressure has to be correlated with the cavitation number (see recommendation no. 1). As a result, the iso-surface with the static pressure is not relevant information to be compared between different operating points. The Q iso-surface can be one option for single phase flow. The iso-surface with gaseous phase can be an option for two-phase flow simulation if the cavitation number and the value of the iso-surface are correlated from one operating point to another.
The internal flow is shown in the draft tube due to the increased swirl intensity in the Fig. 11.
R9: The absolute flows presented in Figure 11 for three guide vane openings are for the student not for a scientific paper. The information about the absolute flow is quantified based on absolute flow angle. Please report the absolute flow angle versus the dimensionless radial coordinate in Figure 12 according to the recommendation no. 7 (R7). This plot gives a view of the absolute flow from the wall to the center of the cone.
It is mentioned on page 14 that “this study confirmed that the inter-blade vortex characteristics in the runner passages depend on the flow rate conditions and velocity components at the runner inlet and the vortex rope characteristics in the draft tube occur in specific flow rate ranges and velocity conditions at the outlet of runner. The flow and velocity characteristics at the outlet of runner are influenced from the internal flow characteristics in the runner passages, and to investigate the correlation of the inter-blade vortex and vortex rope, Fig. 15 compares the iso-surface and 3D streamline flow distributions from the runner passages to the draft tube during the one runner revolution with GVAs of 12.5° and 7°.”
R10: What should identify the reader on Figure 15? Please indicate on each figure what should follow the reader. Please define the inter-blade vortex characteristics. What quantity (quantities) associated to the inter-blade vortex is (are) targeted? Please define this (these) quantity (quantities)? Please assess the distribution of the quantity (quantities) associated to the inter-blade vortex against the volumetric flow rate (discharge). How is influenced the flow in the draft tube by the inter-blade hydrodynamic instabilities? Please provide a quantitative assessment of the influence of the inter-blade instabilities on the hydrodynamic phenomena developed in the draft tube cone (e.g. vortex rope).
The Fourier transform of the unsteady signals acquired on all three monitors (P1, P7 and P14) located between the guide vane and the runner and three monitors (L1, L2 and L3) located on level P2 on the cone wall are plotted in Figures 16 and 17, respectively. The magnitude value is normalized with the maximum value while the frequency is normalized by the rotational frequency (fn) of the pump-turbine.
R11: Each magnitude value should be made dimensionless with the following quantity rho*E where rho [kg/m^3] is the water density and E [J/kg] (E=gH) is the specific energy of the pump-turbine and H [m] is the head associated to the operating point. As a result, the Fourier spectra can be compared for all operating points investigated. The Fourier spectra obtained based on the numerical simulations should be validated against experimental spectra. Usually, the Fourier spectra obtained based on the numerical simulations embeds spurious high harmonics. The validation of the numerical Fourier spectra against experimental ones should be performed to indentify these spurious high harmonics. Then, both synchronous and asynchronous components of the unsteady pressure field should be discriminated based on the signals available using the methodology presented by Bosioc et al. (2012) in order to indentify each type in the power spectrum.
R12: Finally, the authors should clearly state the mechanism of the hydrodynamic instabilities (e.g. vortex rope) develop in the draft tube cone based on their investigations. Several authors (e.g. Krishnamachar P. et al. (2008), Fay (2010)) have stated that the main cause of the hydrodynamic instabilities (e.g. vortex rope) encountered in the draft tube cone is the hydrodynamic instabilities develop in the runner (e.g. rotating stall, inter-blade vortices). Other authors (e.g. Resiga et al. (2006), Pasche et al. (2017)) have promoted the theory that the main cause of the hydrodynamic instabilities (e.g. vortex rope) developed in the draft tube cone is due to the decelerated swirling flows lose their stability. A valuable contribution of the paper should be the elucidation of the mechanism of the hydrodynamic instabilities encountered in the draft tube cone.
Muntean S., Ninaci I., Susan-Resiga R., Baya A., Anton I., Numerical analysis of the flow in the old Francis runner in order to define the refurbishment strategy, UPB Scientific Bulletin, Series D: Mechanical Engineering, 72(1):117 – 124, 2010.
Frunzaverde D., Muntean S., Marginean G., Campian V., Marsavina L., Terzi R., Serban V., Failure analysis of a Francis turbine runner, IOP Conference Series-Earth and Environmental Science, 12, Paper No 012115, 2010.
Susan-Resiga R., Ciocan G.D., Anton I., Avellan F., Analysis of the swirling flow downstream a Francis turbine runner, Journal of Fluids Engineering, 128(1):177-189, 2006.
Bosioc A.I., Susan-Resiga R.F., Muntean S., Tanasa C., Unsteady Pressure Analysis of a Swirling Flow with Vortex Rope and Axial Water Injection in a Discharge Cone, Journal of Fluids Engineering, 134(8):081104, 2012,
Krishnamachar P., Fay A.A., Rangnekar S., Runner core cavitation near full load in Francis turbines, Water Power and Dam Construction, 60(10):36-38, 2008.
Fay A.A., Analysis of low-frequency pulsations in Francis turbines, IOP Conference Series-Earth and Environmental Science, 12, Paper No 012015, 2010.
Pasche S., Avellan F., Gallaire F., Part Load Vortex Rope as a Global Unstable Mode, Journal of Fluids Engineering, 139(5):051102, 2017.
Reviewer 3 Report
1. There are many researches devoted to the interblade vortices in turbines which have common features with the interblade vortices in pump-turbines. In the same time only one reference on such study [5] (among 12) is made in the introduction.
2. The authors performed RANS analyses using a k-ω-based SST turbulence model. There are no discussion why RANS, and k-ω-based SST turbulence model were chosen.
3. In strings 76-77 it is written "Kirschner et al. [11] conducted an experimental investigation with visualization of the dynamic pressure characteristics in the vortex structure ...". How the pressure characteristics can be visualized?
4. In strings 148-149 it is written "the unsteady-state analysis was performed over a total of five revolutions of the runner". The question is how can authors resolve frequency 0.2fn in this case?
5. In strings 149-150 it is written "To improve convergence, the number of iterations was set to five.". What iterations are authors taking about?
6. To validate the numerical analysis results, authors compare the steady- and unsteady-state RANS analysis results and the experimental results. In strings 162-163 they wrote "The unsteady state analysis results were averaged with values during the last revolution of the runner". Well, but talk is on the low flow rate operation conditions. In this case one has precessing vortex rope with frequency much lower then the runner rotation frequency. So, what is meaning of the averaging over one revolution of the runner?
7. What mean abbreviations "POW" and "EFF" in the legend to Figure 5?
8. Iso-surfaces of what characteristic is shown in Figure 6?
9. In strings 224-225 it is written "Figure 10 shows the iso-surface distributions of the time-averaged values of the pressure in the draft tube from the unsteady-state analyses". First question, what pressure level is shown? Second question, as at GVA of 12.5° it is observed the precessing vortex rope, then the time-averaged pressure distribution should be close to axi-symmetrical one. In Figure 10 at GVAs from 8.5° to 12.5° the vortex rope looks like in a fixed moment of time rather than time-averaged. Some comments more regarding this paragraph. The authors say about the size of the vortex rope. The don't know the size of the vortex rope. They can say only about size of the area occupied by the iso-surface. So the phrases "finally the vortex rope disappeared at the GVA of 7°" and "the vortex rope disappeared at 0.30QBEP" look as non-proper. It is possible that with using Q-criterion one will see the vortex rope at this condition. It seems strange that authors analyze the structure of the interblade vortices using a Q-criterion, but to understand the vortex rope structure in the draft tube they consider either a pressure iso-surface or the time-averaged streamlines. So there is doubt on the conclusion that at 0.30QBEP condition the vortex rope did not develop (strings 252 and 318-319).
10. Next questions regards Figure 12. Usually researchers consider the axial and circumferential velocity components in the conical part of the drat tube. Here we see axial and radial velocity components. And the radial component has very high values at GVA of 12.5°. If authors received in calculations such result, the calculations are wrong.
11. There are few misprint or stylistic inaccuracies in the text. "draft wall" instead of "draft tube wall" (string 78). In all places where the term "iso-surface" is used, it should be pointed the characteristic which iso-surface is analyzed. Some word is omitted in string 325 "on the of guide". "black flow" instead of "back flow" (string 396).
Round 2
Reviewer 1 Report
The reviewer found that the quality of the manuscript has been improved. Thus, the manuscript is suitable for publication. The following comments should be addressed to further blush up the manuscript before publication.
(1) Add the original definition of the swirl number S in the text. (Not the analytical one give in the equation 7). It would be helpful to cite the following articles for the definition of the swirl number.
Swirl Flow in Conical Diffuser, 1978, Senoo Y., Kawaguchi N., Nagata T (Bulletin of the JSME, Vol.21) An Outlook on the Draft-Tube-Surge Study, 2013, Nishi M., Liu S. (IJFMS, Vol.6)(2) According to the figure 11, there is a large discrepancy between the analytical Swirl number and the actual Swirl number. It would be interesting to have authors' comments for this.
(3) Swirl number is given by the ratio between the fluxes of the angular momentum and the axial momentum on a given surface. Although the authors provide the distributions of Swirl number on the line, it should be also calculated over the plane (for example, observed plane p3 by surface integral of the fluxes according to the definition given by (1)) and should be compared with the analytical value in Figure 11(a).
Reviewer 2 Report
Several recommendations have been performed in the first review report to improve the scientific level of the manuscript as well as the clarity of the manuscript that should be easily followed by readers. A detailed analysis of the authors’ answers is given in my second review report. Some of recommendations were solved while others were not addressed. The authors define the dimensionless quantities according to the international standard IEC 60193. Then they are using other normalized values to plot the results. The authors have been applied a wrong procedure to obtain the normalized values (e.g. normalized velocity components, normalized flow angles, normalized pressure, normalized amplitude and so on). In this case, the comparison between the data obtained for different operating points is irrelevant. Several details are given in my second review report about this issue (see comments C9 and C11). It was suggested in the first review report to use a relevant quantity for the dimensionless value of the Fourier spectra amplitude. Contrary, the authors selected a maximum local value without any relevance. It was recommended (C12) to quantify the influence of the flow instabilities developed in the runner on the flow instabilities developed in the draft tube cone. This analysis can bring relevant data to elucidate the mechanism of developing the flow instabilities developed in the draft tube cone. The authors did not answered to this basic question although a lot of data is available in their investigations. As a result, I am keeping my previous recommendation MAJOR REVISION for the actual form of the manuscript.
Reviewer 3 Report
In the answer on the comment 4 the authors explain that they resolve frequency 0.2fn due to FFT analysis over five revolutions of the runner. This means that a single period of the frequency 0.2fn was considered. In my opinion such result can't be considered as reliable one. Much longer time interval is required to obtain reliable data.
After answer on the comment 6 I can't understand till now what useful information can be obtained from averaging of the unsteady-state data over one revolution of the runner. If one tries to compare steady- and unsteady-state analysis results and there exists low frequency process (0.2fn), it will be necessary to average at least over one revolution of the vortex.
The same question relates to the comment 9. If one has precessing vortex rope, then at averaging over one revolution of the runner it will be some blurry image of the vortex. Usually researchers show picture of the vortex in few different phases to demonstrate its rotation. The meaning of averaging remains unclear.
Round 3
Reviewer 2 Report
Several recommendations have been performed both in the first and the second review reports to improve the scientific level of the manuscript as well as the clarity of the manuscript. A few of them are taken into account. The scientific level of the manuscript is partially improved. I am recommending to be considered for publication in tha actual form.
Reviewer 3 Report
In their reply on Comment 1 of Review II the authors don’t agree with the comment on the reliability of the frequency determination. To support their arguments, they wrote on “some previous studies were observed with similar range of the frequency from the vortex rope in the draft tube as the frequency of 0.2fn”. Consider for example paper by Favrel et al., Experiments in Fluids, 2015. One can read in the Introduction “… precessing helical vortex core in the draft tube. Its rotational frequency lies between 0.2 and 0.4 times the runner frequency (Nishi et al. 1982).”. There are many papers which confirm this range of frequencies – from 0.2fn till 0.4fn. My comment was on the length of time period taken for determination of this low frequency. In manuscript this period is five revolutions of the runner, i.e. one revolution of the vortex rope. To obtain reliable data on the precession frequency one need in few revolutions of the vortex. Say, 5 revolutions, better 10 revolutions, at least 3 revolutions (statistics over three periods). The authors reply that they compared result obtained over 1 revolution of the vortex with the result obtained over 1.6 revolution of the vortex (8 revolutions of the runner). There is no influence on the reliability, 5 or 8 revolutions. I repeat that much longer time interval is required to obtain reliable data.
In their reply on Comment 2 of Review II the authors wrote “The results of unsteady state analysis show a constant cycle over one revolution of runner with maximum and minimum values”. In the same time in reply on Review I the authors sent a figure with unsteady pressure distribution calculated during five revolutions of the runner. So, every runner revolution we see different pictures. And the phrase “because the frequency from vortex is lower than the rotating frequency of runner, the results over one revolution of runner include at least one revolution of the vortex” killed me finally. If you have frequency of the vortex rope 0.2fn, you see only some 1/5 of the vortex period during one revolution of the runner. The situation is contrary: one revolution of the vortex includes five revolutions of the runner.